# HAIF-GS: Hierarchical and Induced Flow-Guided Gaussian Splatting for Dynamic Scene

**Jianing Chen**[1,2], **Zehao Li**[1,2], **Yujun Cai**[3], **Hao Jiang**[1,2]*, **Chengxuan Qian**[4]
**Juyuan Kang**[1,2], **Shuqin Gao**[1], **Honglong Zhao**[1], **Tianlu Mao**[1,2], **Yucheng Zhang**[1,2]*

[1]Institute of Computing Technology, Chinese Academy of Sciences, ICT
[2]University of Chinese Academy of Sciences, UCAS
[3]The University of Queensland
[4]Jiangsu University
{chenjianing23s, jianghao}@ict.ac.cn

## Abstract

Reconstructing dynamic 3D scenes from monocular videos remains a fundamental challenge in 3D vision. While 3D Gaussian Splatting (3DGS) achieves real-time rendering in static settings, extending it to dynamic scenes is challenging due to the difficulty of learning structured and temporally consistent motion representations. This challenge often manifests as three limitations in existing methods: redundant Gaussian updates, insufficient motion supervision, and weak modeling of complex non-rigid deformations. These issues collectively hinder coherent and efficient dynamic reconstruction. To address these limitations, we propose **HAIF-GS**, a unified framework that enables structured and consistent dynamic modeling through sparse anchor-driven deformation. It first identifies motion-relevant regions via an Anchor Filter to suppress redundant updates in static areas. A self-supervised Induced Flow-Guided Deformation module induces anchor motion using multi-frame feature aggregation, eliminating the need for explicit flow labels. To further handle fine-grained deformations, a Hierarchical Anchor Propagation mechanism increases anchor resolution based on motion complexity and propagates multi-level transformations. Extensive experiments on synthetic and real-world benchmarks validate that HAIF-GS significantly outperforms prior dynamic 3DGS methods in rendering quality, temporal coherence, and reconstruction efficiency.

## 1 Introduction

Reconstructing dynamic 3D scenes from monocular video represents a fundamental challenge in computer vision with applications spanning virtual reality [13, 15] and autonomous driving [4, 41, 48]. Unlike static reconstruction, dynamic environments introduce complex deformations time-varying geometry, significantly complicating consistent modeling of both appearance and motion.

While Neural Radiance Fields (NeRF) [27] revolutionized static scene representation, their implicit volumetric nature requires dense sampling, resulting in prohibitive computational costs and slow rendering. 3D Gaussian Splatting (3DGS) [14] has emerged as an efficient alternative, replacing volume integration with explicit 3D Gaussians for high visual fidelity and real-time rendering. However, 3DGS was designed for static scenes, and extending it to dynamic ones remains challenging.

Several recent methods [44, 39, 40, 49] extend 3DGS to dynamic scenes by introducing learned deformation fields. These approaches typically employ MLPs to predict time-dependent transformations of Gaussian parameters (*e.g.* position, quaternion). While effective in capturing motion, they

---

*Corresponding authors

39th Conference on Neural Information Processing Systems (NeurIPS 2025).

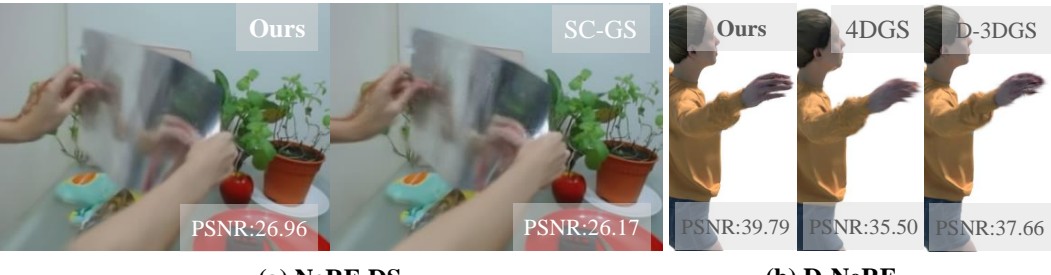

**(a) NeRF-DS**           **(b) D-NeRF**

Figure 1. The visualization results on (a) NeRF-DS [42] dataset and (b) D-NeRF [31] dataset.

require querying and updating an excessive number of Gaussians at each timestep, which leads to considerable redundancy. To improve efficiency, subsequent methods [12, 5] introduce a sparse set of control points to derive transformations of nearby Gaussians via interpolation. However, such strategies often fail to capture fine-grained or non-rigid motion. For example, Figure 1 (b) shows hand motion, where the fine-scale appearance and localized movement highlight the challenges of modeling fine-grained dynamics in dynamic reconstruction. This limitation is mainly due to two factors: (1) Limited expressiveness: Deformation fields typically employ simple MLPs that lack the capacity to model articulated or spatially varying motion, particularly in complex real-world scenes. (2) Insufficient motion supervision: Training relies solely on image reconstruction loss, without explicit motion guidance or structural constraints, resulting in temporal inconsistencies and artifacts.

To address this challenge, we propose HAIF-GS (Hierarchically Anchored and Induced Flow-Guided Gaussian Splatting), a unified framework that significantly improves both the efficiency and accuracy of dynamic scene reconstruction. HAIF-GS builds upon sparse motion anchors as the core deformation units, substantially reducing redundant queries over individual Gaussians and forming a compact foundation for anchor-driven motion modeling. Building on this foundation, we introduce three complementary modules to further improve the quality of deformation: (1) Unlike prior methods that process the transformation of all Gaussians indiscriminately through deformation fields, we employ an Anchor Filter module that predicts a dynamic confidence score for each anchor, allowing the model to select motion-relevant anchors and concentrate deformation learning on dynamic regions. (2) We introduce the Induced Flow-Guided Deformation (IFGD) module, which leverages multi-frame aggregation to induce scene flow in a self-supervised manner and uses it to regularize anchor transformations without requiring external flow labels. This design enables more structured supervision for motion learning and improves the temporal consistency of anchor trajectories. (3) To further enhance motion expressiveness, we employ the Hierarchical Anchors Densification (HAD) module, which progressively densifies anchors in motion-complex regions and establishes a layered structure for hierarchical motion propagation. Assigning new anchors to a parent from upper layers allows the model to transfer transformation cues across levels and to capture fine-grained and non-rigid deformations in challenging regions. By integrating these techniques, our HAIF-GS achieves efficient and accurate reconstruction of dynamic scenes with diverse and complex motion patterns, benefiting from the synergy between sparse representation, targeted supervision, and hierarchical motion modeling.

In summary, our contributions are as follows:

- We propose HAIF-GS, a unified framework for dynamic scene reconstruction based on sparse motion anchors to reduce redundancy and improve efficiency, while addressing key challenges of temporal inconsistency and local non-rigidity.

- We introduce a motion modeling strategy that integrates motion-aware anchor filtering, induced flow-guided deformation, and hierarchical anchor densification, enabling fine-grained, temporally consistent, and structurally adaptive representation of complex dynamics.

- Extensive experiments on both real-world (NeRF-DS) and synthetic (D-NeRF) datasets demonstrate that HAIF-GS achieves state-of-the-art performance quantitatively and qualitatively.

## 2 Related Work

**Dynamic NeRF.** Neural Radiance Fields (NeRF) [27] have become a powerful framework for novel view synthesis in static scenes by implicitly modeling volumetric radiance with MLPs, offering an alternative to traditional multi-view stereo (MVS) approaches [45, 46]. A broad range of works [11, 9, 29–31, 7, 43, 36] have extended NeRF to dynamic settings by introducing temporal representations such as deformation fields and canonical mappings. Despite their effectiveness, these methods remain inefficient due to dense ray sampling and costly volume rendering. To address this challenge, numerous improved methods [20, 24, 22, 23, 26, 37] have been proposed for dynamic scene rendering, leveraging grid structures [24] and multi-view supervision [22, 23] to enhance both efficiency and reconstruction quality. Recent works accelerate dynamic modeling using explicit representations such as multi-plane [3, 8, 33] and grid-plane hybrids [34], which factorize spatiotemporal space but still render slowly. While these methods significantly accelerate training and inference, their performance remains insufficient for real-time applications.

**Dynamic Gaussian Splatting.** Recently, 3D Gaussian Splatting (3DGS) [14] has gained attention for achieving real-time rendering with explicit point-based representations, and has been increasingly adopted in dynamic reconstruction [44] and has shown potential for broader 3D tasks [18, 32, 1, 2]. Several works have extended 3DGS to dynamic scenes by learning time-varying transformations of Gaussians [44, 39, 40, 12, 5, 25, 49, 19]. Early approaches [44] rely on per-Gaussian deformation fields, which often introduce redundant computations and training inefficiencies. To address this, later methods utilize compact structures such as plane encodings [39] or hash-based representations [40] to model deformation more efficiently. Another line of work [12, 5] employs sparse control points to drive Gaussian motion through interpolation, enabling both high-quality rendering and motion editing. However, existing approaches still struggle to model fine-grained and temporally consistent deformations in a compact and generalizable manner. In contrast, our proposed **HAIF-GS** introduces sparse anchor-based motion modeling with self-supervised flow induction and hierarchical propagation, achieving a better balance between efficiency and expressive motion representation.

## 3 Preliminary

Before introducing our approach, we briefly review key background concepts, including the 3DGS for static scenes (Section 3.1) and its recent dynamic extensions (Section 3.2).

### 3.1 3D Gaussian Splatting

3D Gaussian Splatting (3DGS) [14] has emerged as a powerful technique for real-time rendering of static scenes. In this representation, the scene is modeled as collections of anisotropic 3D Gaussians, each defined by a set of continuous attributes: center position $\mu \in \mathbb{R}^3$, covariance matrix $\Sigma \in \mathbb{R}^{3 \times 3}$, opacity $\sigma$ and spherical harmonics (SH) coefficients $h$ for view-dependent appearance modeling.

To enable differentiable rendering, the covariance matrix is parameterized via a rotation-scaling decomposition: $\Sigma = RSS^T R^T$, where $R$ is derived from a unit quaternion and $S$ is a diagonal scaling matrix. Each 3D Gaussian projects onto the image plane through a camera projection matrix $W$ and a local Jacobian $J$, creating a 2D elliptical footprint with transformed covariance $\Sigma' = JW\Sigma W^T J^T$ and projected center $\mu' = JW\mu$.

Rendering is performed via a forward splatting process, where Gaussians contribute to nearby pixels via an $\alpha$-blending-based soft visibility weighting scheme. The color at pixel $u$ is computed by aggregating radiance contributions from overlapping Gaussians as:

$$C(u) = \sum_{i \in \mathcal{N}(u)} T_i \alpha_i \text{SH}(h_i, \mathbf{v}), \tag{1}$$

where $T_i = \prod_{j<i}(1 - \alpha_j)$ represents transmittance, $\alpha_i$ is derived from the projected Gaussian density, and $\text{SH}(h_i, \mathbf{v})$ evaluates the spherical harmonic function along view direction $\mathbf{v}$.

The Gaussian parameters are optimized via gradient descent using a photometric reconstruction loss. Adaptive density control is applied to prune redundant Gaussians and spawn new ones based on the optimization signal, improving convergence and rendering quality.

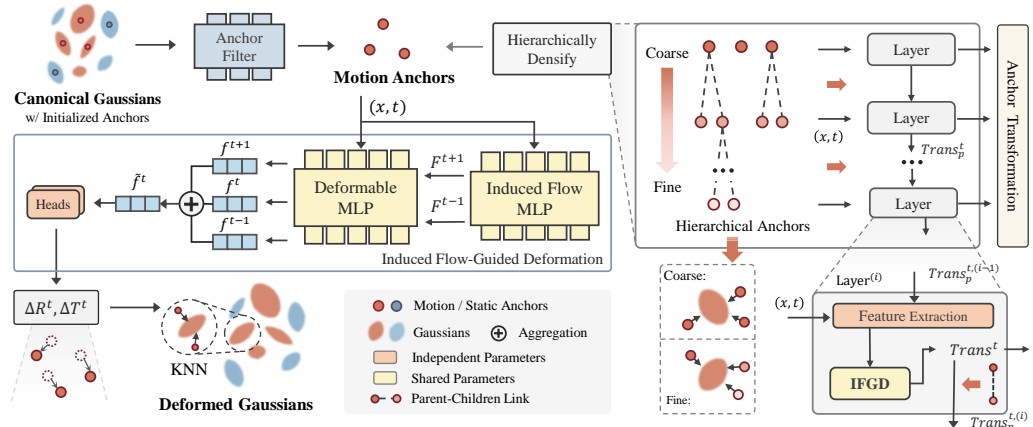

Figure 2. **The overview of our HAIF-GS.** Given the canonical Gaussians, we first initialize sparse motion anchors and filter them using a confidence-aware Anchor Filter. We then aggregate multi-frame features to predict temporally consistent transformations for each anchor. In regions with complex motion, we hierarchically densify anchors and propagate deformations across layers. Finally, we update Gaussian parameters via anchor-based interpolation and render images for supervision.

## 3.2 Dynamic Reconstruction Based on 3DGS

While 3DGS excels at static scene, extending it to dynamic scenes presents challenges. Recent methods address this issue by extending 3DGS to dynamic settings through explicit motion modeling in the Gaussian representation. A common approach learns a deformation field that transforms static Gaussians over time. Specifically, a deformation network $D$ inputs a Gaussian's canonical attributes (center position $\mu$, rotation $r$, scaling $s$) along with timestamp $t$, and outputs temporal offsets:

$$(\mu_t, r_t, s_t) = D(\mu, r, s, t), \tag{2}$$

where $(\mu_t, r_t, s_t)$ denote motion-compensated attributes at time $t$, enabling the modeling of non-rigid deformations across frames for dynamic scene reconstruction.

In practice, deformable 3DGS can be further enhanced by combining motion priors, camera pose refinement, and feature-guided correspondence mechanisms. However, existing approaches often struggle to achieve efficient modeling of complex local motion, maintain temporal consistency across frames, and capture heterogeneous motion patterns in real-world scenarios. These limitations motivate the development of more structured and scalable motion representations for dynamic reconstruction.

## 4 Method

### 4.1 Overview

Given image sequences $\{I_t\}$ and corresponding camera poses $\{P_t\}$ at timestamps $\{t\}$, our goal is to learn a representation that can render the dynamic scene from arbitrary viewpoints at any time within the observed interval. This task presents inherent challenges: efficient updates of numerous Gaussians across time; heterogeneous motion patterns in real-world scenes combining static backgrounds with complex deformations; and modeling detailed non-rigid motion with limited viewpoints.

Our approach addresses these challenges through a hierarchical sparse motion representation based on 3D Gaussian Splatting, as illustrated in Figure 2. We represent motion using a sparse set of learnable 3D motion anchors that predict local transformations propagated to Gaussians through spatial interpolation. To enhance temporal consistency and motion prediction, we introduce an induced flow-guided deformation module that aggregates multi-frame features to guide anchor transformations without explicit flow supervision. For complex motion regions, we introduce a hierarchical anchor structure that refines predictions with denser anchors at higher resolution. Our method comprises three components: sparse motion anchors and dynamic decomposition (Section 4.2), induced flow-guided deformation (Section 4.3), and Hierarchical Anchor Densification (Section 4.4).

## 4.2 Sparse Motion Anchors and Dynamic Decomposition

**Motion anchor representation.** We represent scene motion using a sparse set of learnable 3D points, referred to as motion anchors, which serve as the primary units for modeling deformation. Instead of directly associating motion parameters with each Gaussian, we predict local transformations from anchors and propagate them to Gaussian through spatial interpolation.

Motion anchors are initialized via farthest point sampling (FPS) in the canonical space to ensure uniform spatial coverage. Each anchor $a_i$ is parameterized by a 3D position $x_i \in \mathbb{R}^3$ and a learnable influence scale $\rho_i \in \mathbb{R}^+$. The set of all anchors is denoted as $\mathcal{A} = \{(x_i, \rho_i)\}_{i=1}^M$.

For each Gaussian $g_j$ centered at $\mu_j$, we select its $K$ nearest motion anchors $\mathcal{A}_j$ based on Euclidean distance in canonical space. The influence of each anchor on a specific Gaussian is determined by a normalized Gaussian kernel [6, 28]:

$$\omega_{ij} = \frac{\exp\left(-\|\mu_j - x_i\|^2/\rho_i^2\right)}{\sum_{x_k \in \mathcal{A}_j} \exp\left(-\|\mu_j - x_k\|^2/\rho_k^2\right)}. \tag{3}$$

The spatial interpolation weights $\omega_{ij}$ are used to aggregate transformations from neighboring anchors to compute the final motion of each Gaussian.

**Dynamic-Static decomposition.** To distinguish between dynamic and static motion patterns, we introduce a decomposition strategy over the set of motion anchors. Specifically, we assign to each anchor $a_i$ a confidence score $\alpha_i \in [0, 1]$ that reflects the likelihood of the anchor participating in dynamic motion. Anchors with high confidence are treated as dynamic and contribute to deformation modeling, while static anchors are excluded.

Each $\alpha_i$ is predicted by an Anchor Filter, a lightweight MLP that takes as input the position encoding of the anchor's spatial and temporal coordinates $\gamma(x_i), \gamma(t)$. The Anchor Filter is jointly optimized with the the Induced Flow-Guided Deformation module to enable end-to-end training. During training, we adopt a two-stage scheme: in the first stage, all anchors contribute to the deformation field with their predicted transformations modulated by the soft weight $\alpha_i$, allowing gradients to propagate to the Anchor Filter; in the second stage, a hard decomposition is applied by thresholding $\alpha_i$, and only dynamic anchors are used for motion modeling.

To regularize the dynamic-static decomposition, we introduce two regularization losses that encourage confident and sparse selection of motion anchors. The details are described in Section 4.5.

## 4.3 Induced Flow-Guided Deformation

To leverage temporal context, we introduce the Induced Flow-Guided Deformation module, which consists of two key components. The Induced Flow MLP predicts forward and backward scene flows from anchor position and timestamp after position encoding, providing three temporally shifted queries across adjacent frames. The Deformable MLP maps each flow-guided query to a feature embedding and then integrates them into a temporally aligned feature, which is used to predict anchor-wise transformations (*i.e.* rotation and translation) through multiple transformation heads. The induced flows serve as internal guidance signals, facilitating multi-frame feature alignment and thereby promoting temporally consistent motion modeling. Notably, both the dynamic-static separation and flow-based motion learning are induced during scene reconstruction optimization, without requiring external flow supervision.

**Induced flow-guided temporal queries and feature aggregation.** Given a motion anchor $x$ and time $t$, after positional encoding, the Induced Flow MLP predicts backward and forward scene flow:

$$(\boldsymbol{F}^{t-1}, \boldsymbol{F}^{t+1}) = \text{MLP}_{\text{flow}}(x, t),$$

where $\boldsymbol{F}^{t-1}$ and $\boldsymbol{F}^{t+1}$ represent the anchor displacement from $t$ to $t-1$ and $t+1$, respectively. These flows guide the construction of three temporally shifted queries across adjacent frames:

$$q_{t-1} = (x + \boldsymbol{F}^{t-1},\ t-1), \quad q_t = (x,\ t), \quad q_{t+1} = (x + \boldsymbol{F}^{t+1},\ t+1).$$

Each flow-guided query is mapped to a time-aware feature embedding by a shared Deformable MLP:

$$\boldsymbol{f}^{t+i} = \text{MLP}_{\text{deform}}(q_{t+i}), \quad i \in \{-1, 0, +1\}.$$

These embeddings are aggregated using a simple weight to produce a temporally consistent feature:

$$\tilde{f}^t = \lambda f^{t-1} + (1 - 2\lambda) f^t + \lambda f^{t+1}, \tag{4}$$

where $\lambda \in (0, 0.5)$ is a fusion weight that emphasizes the reference frame while incorporating motion context from neighboring frames. We set $\lambda = 0.25$ in all experiments.

This temporal aggregation bridges scene flow prediction and scene reconstruction, enabling their joint optimization during training. Embedding temporal consistency into anchor representations enables accurate prediction by sharing information across frames. The flow fields are learned without explicit flow supervision; instead, multi-frame feature aggregation implicitly *induces* flow prediction to converge toward coherent and consistent motion patterns during scene reconstruction.

**Transformation prediction and interpolation.** The temporally consistent feature $\tilde{f}_i^t$ of anchor $a_i$ at timestep $t$ is used to predict its translation $\Delta T_i^t \in \mathbb{R}^3$ and rotation $\Delta R_i^t \in \mathbb{R}^3$ through multiple independent transformation heads. Given these predicted anchor transformations, our next goal is to drive the motion of individual Gaussian using nearby anchors. For each Gaussian $j$, we identify a set of neighboring motion anchors $\mathcal{A}_j$ via K-nearest neighbor (KNN) search in the canonical space. The corresponding spatial interpolation weights $\omega_{ij}$ are computed based on the spatial distance between Gaussian centers and anchors, as defined in Eq. 3. Following the prior dynamic fusion methods [12, 6], we adopt the linear blend skinning (LBS) strategy [35] to compute the Gaussian deformation $\Delta x_j^t, \Delta r_j^t$ by interpolating the transformations of its neighboring anchors:

$$\Delta \mu_j^t = \sum_{i \in \mathcal{A}_j} \omega_{ij} \left( \Delta R_i^t (\mu_j - x_i) + x_i + \Delta T_i^t \right), \quad \Delta r_j^t = \left( \sum_{i \in \mathcal{A}_j} \omega_{ij} \cdot \Delta R_i^t \right) \otimes r_j, \tag{5}$$

where $\mu_j$ and $x_i$ denote the centers of Gaussian $j$ and anchor $i$, and $\otimes$ is quaternion multiplication. This interpolation allows each Gaussian's deformation to be locally driven by a sparse anchor transformations, enabling compact modeling of dense deformation fields.

**Temporal self-supervision.** To encourage temporal consistency in motion learning, we apply a cycle consistency loss on the predicted scene flow. This loss is described in detail in Section 4.5.

## 4.4 Hierarchical Anchor Densification

While sparse motion anchors efficiently model global motion, they may struggle to capture fine-grained, non-rigid deformations. We address this with a hierarchical anchor structure that introduces finer-resolution anchors in regions of high motion complexity.

We identify regions requiring finer-resolution anchors by computing the temporal translation variance for each base-level anchor. Specifically, for each anchor $a_i$, we compute the normalized variance $var(a_i)$ of its predicted translation $\Delta T_i$ across $N_t (= 16)$ randomly sampled timestamps as:

$$var(a_i) = \frac{1}{N_t} \sum_{t=1}^{N_t} \left\| \Delta T_i^t - \overline{\Delta T_i} \right\|_2^2, \quad \text{where} \quad \overline{\Delta T_i} = \frac{1}{N_t} \sum_{t=1}^{N_t} \Delta T_i^t. \tag{6}$$

Anchors with variance $var$ exceeding a threshold $\tau$ are marked for hierarchical refinement. We generate finer-resolution anchors by duplicating each selected anchor with small spatial offsets, forming a multi-scale hierarchy that captures progressively finer motion details.

Figure 2 illustrates our hierarchical anchor densification module, which refines motion representations across multiple anchor layers in a coarse-to-fine manner. Each child anchor encodes its own spatial position, timestamp, and the parent translation, enabling cross-level motion propagation. All layers share the deformation MLP while using lightweight, layer-specific feature extractors for efficiency.

For each Gaussian's final transformation, we independently retrieve neighboring anchors from each layer, apply weighted interpolation within layers, and fuse the results using level-specific learned confidence weights. This enables multi-scale motion decomposition with minimal computational overhead, as only regions requiring fine-grained modeling incur the cost of additional anchor levels.

Table 1. **Quantitative comparison on NeRF-DS dataset per-scene.** We highlight the best , second best and the third best results in each scene. The rendering resolution is set to $480 \times 270$. NeRF-DS, HyperNeRF, 4DGS, SC-GS and ours use MS-SSIM/LPIPS (Alex), while other methods employ SSIM/LPIPS (VGG).

| Method | Sieve | | | Plate | | | Bell | | | Press | | |
|---|---|---|---|---|---|---|---|---|---|---|---|---|
| | PSNR↑ | MS-SSIM↑ | LPIPS↓ | PSNR↑ | MS-SSIM↑ | LPIPS↓ | PSNR↑ | MS-SSIM↑ | LPIPS↓ | PSNR↑ | MS-SSIM↑ | LPIPS↓ |
| 3DGS [14] | 23.16 | 0.8203 | 0.2247 | 16.14 | 0.6970 | 0.4093 | 21.01 | 0.7885 | 0.2503 | 22.89 | 0.8163 | 0.2904 |
| HyperNeRF [30] | 25.43 | 0.8798 | 0.1645 | 18.93 | 0.7709 | 0.2940 | 23.06 | 0.8097 | 0.2052 | 26.15 | 0.8897 | 0.1959 |
| NeRF-DS [42] | 25.78 | 0.8900 | 0.1472 | 20.54 | 0.8042 | 0.1996 | 23.19 | 0.8212 | 0.1867 | 25.72 | 0.8618 | 0.2047 |
| 4DGS [39] | 26.11 | 0.9193 | 0.1107 | 20.41 | 0.8311 | 0.2010 | 25.70 | 0.9088 | 0.1103 | 26.72 | 0.9031 | 0.1301 |
| Deformable 3DGS [44] | 25.70 | 0.8715 | 0.1504 | 20.48 | 0.8124 | 0.2224 | 25.74 | 0.8503 | 0.1537 | 26.01 | 0.8646 | 0.1905 |
| SC-GS [12] | 25.93 | 0.9187 | 0.1194 | 20.17 | 0.8257 | 0.2104 | 25.97 | 0.9172 | 0.1167 | 26.57 | 0.8971 | 0.1367 |
| Ours | 26.61 | 0.9374 | 0.0933 | 20.96 | 0.8379 | 0.2181 | 26.34 | 0.9352 | 0.1162 | 27.05 | 0.9133 | 0.1259 |

| Method | Cup | | | As | | | Basin | | | Mean | | |
|---|---|---|---|---|---|---|---|---|---|---|---|---|
| | PSNR↑ | MS-SSIM↑ | LPIPS↓ | PSNR↑ | MS-SSIM↑ | LPIPS↓ | PSNR↑ | MS-SSIM↑ | LPIPS↓ | PSNR↑ | MS-SSIM↑ | LPIPS↓ |
| 3DGS [14] | 21.71 | 0.8304 | 0.2548 | 22.69 | 0.8017 | 0.2994 | 18.42 | 0.7170 | 0.3153 | 20.29 | 0.7816 | 0.2920 |
| HyperNeRF [30] | 24.59 | 0.8770 | 0.1650 | 25.58 | 0.8949 | 0.1777 | 20.41 | 0.8199 | 0.1911 | 23.45 | 0.8488 | 0.1990 |
| NeRF-DS [42] | 24.91 | 0.8741 | 0.1737 | 25.13 | 0.8778 | 0.1741 | 19.96 | 0.8166 | 0.1855 | 23.60 | 0.8494 | 0.1816 |
| 4DGS [39] | 24.57 | 0.9102 | 0.1185 | 26.30 | 0.8917 | 0.1499 | 19.01 | 0.8277 | 0.1631 | 24.18 | 0.8845 | 0.1405 |
| Deformable 3DGS [44] | 24.86 | 0.8908 | 0.1532 | 26.31 | 0.8842 | 0.1783 | 19.67 | 0.7934 | 0.1901 | 24.11 | 0.8524 | 0.1769 |
| SC-GS [12] | 24.32 | 0.9121 | 0.1207 | 26.17 | 0.8851 | 0.1491 | 19.23 | 0.8379 | 0.1514 | 24.05 | 0.8848 | 0.1439 |
| Ours | 24.72 | 0.9255 | 0.1124 | 26.96 | 0.9048 | 0.1247 | 19.74 | 0.8593 | 0.1488 | 24.63 | 0.9014 | 0.1342 |

## 4.5 Optimization

Although our proposed framework can effectively predict Gaussian deformation through anchor-based modeling, it still faces challenges in maintaining flow coherence and controlling the sparsity of motion anchors. To address these challenges, we introduce three additional loss functions.

The cycle consistency loss is specifically designed to enhance bidirectional consistency by encouraging the forward flow followed by backward flow to return to the original position, in accordance with the physical constraint that holds for most real-world motions:

$$\mathcal{L}_{\text{cycle}} = \mathbb{E}_i[\left\| \boldsymbol{F}_i^{t-1} + \text{Flow}(x_i + \boldsymbol{F}_i^{t-1}, t-1) \right\|_2^2 + \left\| \boldsymbol{F}_i^{t+1} + \text{Flow}(x_i + \boldsymbol{F}_i^{t+1}, t+1) \right\|_2^2], \quad (7)$$

where $Flow$ is the induced flow MLP defined in Section 4.3. This self-supervision signal is crucial for learning coherent motion patterns without requiring explicit motion annotations.

We also apply two regularization losses on the motion confidence scores $\alpha_i$ to regularize the dynamic anchors selected by Anchor Filter:

$$\mathcal{L}_{\text{sparsity}} = \mathbb{E}_i\left[\alpha_i\right], \quad \mathcal{L}_{\text{entropy}} = \mathbb{E}_i\left[\alpha_i(1 - \alpha_i)\right]. \quad (8)$$

The sparsity loss $\mathcal{L}_{\text{sparsity}}$ controls the proportion of dynamic anchors, encouraging the model to explain the scene with as few dynamic elements as possible, following Occam's razor principle that simpler explanations should be preferred. The entropy loss $\mathcal{L}_{\text{entropy}}$ promotes confident binary decisions in the Anchor Filter by penalizing uncertain scores near 0.5.

In general, our final loss function is formulated as a weighted sum of the L1 color loss, the D-SSIM loss, and the three proposed regularization terms, where the photometric losses (L1 and D-SSIM) are similar to those in 3DGS [14] and ensure visual fidelity of the reconstructed scene.

$$\mathcal{L} = \lambda\mathcal{L}_1 + (1 - \lambda)\mathcal{L}_{\text{D-SSIM}} + \lambda_1\mathcal{L}_{\text{cycle}} + \lambda_2\mathcal{L}_{\text{entropy}} + \lambda_3\mathcal{L}_{\text{sparsity}}, \quad (9)$$

where $\lambda$, $\lambda_1$, $\lambda_2$, and $\lambda_3$ are weighting coefficients balancing the relative contributions of each loss term. We empirically set $\lambda$=0.8, $\lambda_1 = 0.01$, $\lambda_2 = 0.2$, and $\lambda_3 = 0.5$ to prioritize reconstruction quality while ensuring temporal consistency and model parsimony.

# 5 Experiments

## 5.1 Experimental Setup

**Datasets and Metrics.** We evaluate our method on two widely used benchmarks for monocular dynamic scene reconstruction: NeRF-DS [42] and D-NeRF [31]. For quantitative evaluation, we

Table 2. **Quantitative comparison on D-NeRF dataset per-scene.** We highlight the the best , second best and the third best results in each scene. The rendering resolution is set to $800 \times 800$.

| Method | Hook | | | Jumping Jacks | | | Trex | | | Bouncing Balls | | |
|---|---|---|---|---|---|---|---|---|---|---|---|---|
| | PSNR↑ | SSIM↑ | LPIPS↓ | PSNR↑ | SSIM↑ | LPIPS↓ | PSNR↑ | SSIM↑ | LPIPS↓ | PSNR↑ | SSIM↑ | LPIPS↓ |
| TiNeuVox [7] | 30.51 | 0.959 | 0.060 | 33.46 | 0.977 | 0.041 | 31.43 | 0.967 | 0.047 | 40.28 | 0.992 | 0.042 |
| 4DGS [39] | 32.95 | 0.977 | 0.027 | 35.50 | 0.986 | 0.020 | 33.95 | 0.985 | 0.022 | 40.77 | 0.994 | 0.015 |
| Deformable 3DGS [44] | 37.06 | 0.986 | 0.016 | 37.66 | 0.989 | 0.013 | 37.56 | 0.993 | 0.010 | 40.91 | 0.995 | 0.009 |
| SC-GS [12] | 38.79 | 0.990 | 0.011 | 39.34 | 0.992 | 0.008 | 39.53 | 0.994 | 0.009 | 41.59 | 0.995 | 0.009 |
| Ours | 39.38 | 0.996 | 0.013 | 39.79 | 0.997 | 0.010 | 40.27 | 0.998 | 0.009 | 41.63 | 0.996 | 0.009 |

| Method | Hell Warrior | | | Mutant | | | Standup | | | Mean | | |
|---|---|---|---|---|---|---|---|---|---|---|---|---|
| | PSNR↑ | SSIM↑ | LPIPS↓ | PSNR↑ | SSIM↑ | LPIPS↓ | PSNR↑ | SSIM↑ | LPIPS↓ | PSNR↑ | SSIM↑ | LPIPS↓ |
| TiNeuVox [7] | 27.29 | 0.964 | 0.076 | 32.07 | 0.961 | 0.048 | 34.46 | 0.980 | 0.033 | 31.92 | 0.972 | 0.038 |
| 4DGS [39] | 28.80 | 0.974 | 0.037 | 37.75 | 0.988 | 0.016 | 38.15 | 0.990 | 0.014 | 35.41 | 0.985 | 0.021 |
| Deformable 3DGS [44] | 41.34 | 0.987 | 0.024 | 42.47 | 0.995 | 0.005 | 44.14 | 0.995 | 0.007 | 40.16 | 0.991 | 0.012 |
| SC-GS [12] | 42.19 | 0.989 | 0.019 | 43.43 | 0.996 | 0.005 | 46.72 | 0.997 | 0.004 | 41.65 | 0.993 | 0.009 |
| Ours | 42.50 | 0.993 | 0.021 | 43.65 | 0.998 | 0.005 | 46.83 | 0.999 | 0.004 | 42.00 | 0.997 | 0.010 |

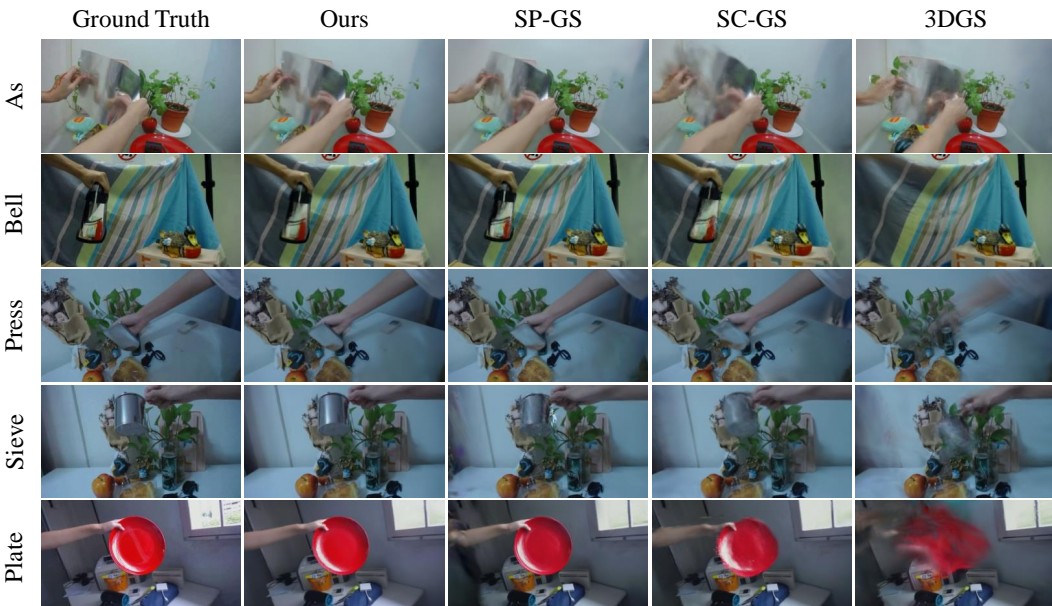

Figure 3. **Qualitative comparison on the NeRF-DS dataset [42]**. Compared with other SOTA methods, our method reconstructs finer details and produces a structured rendering of the moving objects.

employ three standard metrics: Peak Signal-to-Noise Ratio (PSNR), Structural Similarity Index (SSIM), and Learned Perceptual Image Patch Similarity (LPIPS) [47].

**Baselines and Implementation.** We compare our method with state-of-the-art methods in dynamic scene reconstruction, including NeRF-based methods (TiNeuVox [7], HyperNeRF [30], and NeRF-DS [42]) and 3DGS-based methods (3DGS [14], Deformable 3DGS [44], 4DGS [39], SC-GS [12], and SP-GS [5]). All implementations are based on PyTorch framework and trained on a single NVIDIA RTX 3090 GPU. For more implementation details, please refer to Appendix.

## 5.2 Comparisons

**NeRF-DS Dataset.** We evaluate our method on the real-world NeRF-DS dataset. As shown in Table 1, our method outperforms state-of-the-art baselines across all scenes and metrics. Figure 3 shows that our method captures accurate positions and sharp edges of moving objects in scene like *press* and *sieve*, while reducing visual distortions, benefiting from the temporal consistency provided by the guidance of induced flow. Moreover, in the *as* scene, our hierarchical anchor design better models the non-rigid deformation of the moving sheet compared to the single-layer SC-GS, preserving fine-scale structure and spatial coherence.

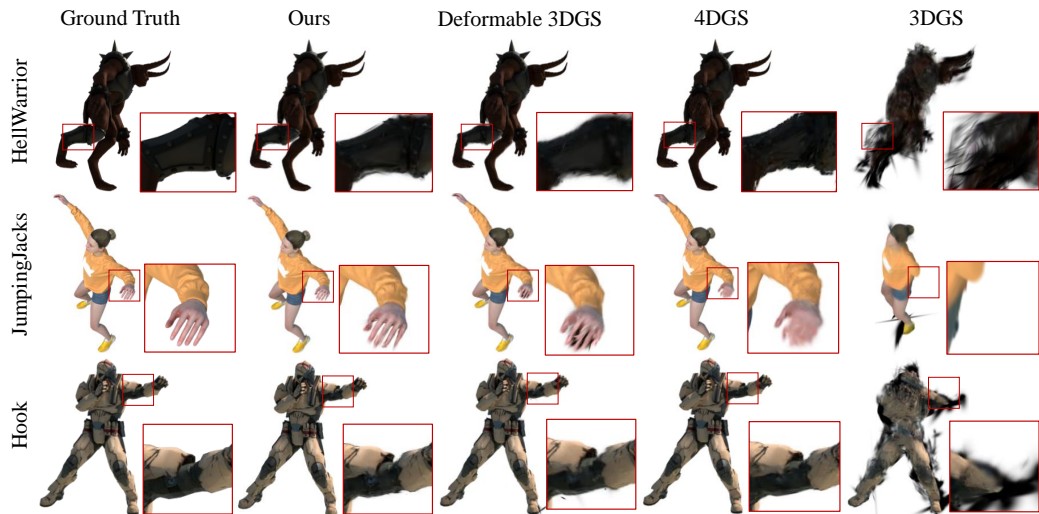

Figure 4. **Qualitative comparison on the D-NeRF dataset [31]**.

Table 3. Ablations on the key components of our proposed framework on NeRF-DS dataset [42].

| Method | PSNR ↑ | MS-SSIM ↑ | LPIPS ↓ |
|---|---|---|---|
| w/o IF | 23.91 | 0.8629 | 0.1624 |
| w/o $\mathcal{L}_{cycle}$ | 24.51 | 0.8947 | 0.1437 |
| w/o AF | 24.45 | 0.8802 | 0.1488 |
| w/o HAD | 24.39 | 0.8774 | 0.1493 |
| Ours | **24.63** | **0.9014** | **0.1342** |

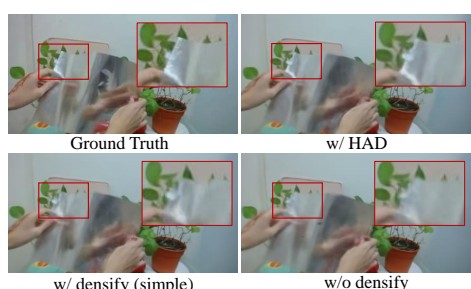

Figure 5. Visualization of anchor densify methods.

**D-NeRF Datasets.** We further validate our method on the synthetic D-NeRF benchmark. The quantitative results summarized in Table 2 show that our approach also achieves the best performance across all evaluated scenes. Figure 4 provides qualitative comparisons, where our method consistently produces more accurate geometry with few artifacts. For example, in hell warrior and jumping jacks scene, our approach better handles body deformation, and recovers coherent limb structures that are distorted or blurred in other methods. These results highlight the effectiveness of our anchor-driven deformation and hierarchical refinement in preserving temporal fidelity and spatial precision.

## 5.3 Ablation Study

**Effectiveness of Induced Flow-Guided Deformation.** We ablate the Induced Flow-Guided Deformation module by removing its two core components: the Induced Flow MLP (IF) and the Cycle Consistency Loss ($\mathcal{L}_{cycle}$). As shown in Table 3 (row 1), removing the entire Induced Flow module leads to a noticeable drop across all metrics, indicating the importance of multi-frame aggregation for temporal consistency. Disabling $\mathcal{L}$cycle (row 2) slightly reduces PSNR and SSIM, showing its role in guiding consistent motion learning.

**Effectiveness of Anchor Filter and Hierarchical Motion Propagation.** Removing the Anchor Filter (AF) causes moderate drops (row 3), confirming its role in suppressing redundancy. Disabling Hierarchical Anchor Densification (HAD) further reduces performance(row 4), showing its effectiveness in modeling complex motions. Figure 5 compares different densification methods, where our hierarchical design better preserves fine-scale deformation than simple or no densification.

# 6 Conclusion

We presented HAIF-GS, a unified framework for dynamic 3D reconstruction that combines sparse anchor-driven modeling with flow-guided deformation and hierarchical motion propagation. By leveraging sparse motion anchors, our method reduces redundant updates and focuses modeling capacity on dynamic regions. The flow-guided component induces temporally consistent anchor transformations through multi-frame aggregation, while the hierarchical propagation module enhances deformation expressiveness in motion-complex areas. Our design addresses key limitations of prior methods by improving temporal consistency, reducing redundancy, and capturing non-rigid motion more effectively. Extensive experiments on both synthetic and real-world benchmarks demonstrate that HAIF-GS achieves superior rendering quality, motion fidelity, and efficiency, advancing the state of the art in dynamic scene reconstruction.

**Limitations.** While HAIF-GS demonstrates strong performance across diverse dynamic scenes, it still exhibits limitations. First, although the anchor filtering and hierarchical propagation reduce redundancy, the overall memory footprint remains non-trivial when modeling scenes with highly complex motion dynamics due to anchor proliferation. Second, the self-supervised scene flow is induced implicitly via multi-frame feature alignment, which may be sensitive to significant occlusions regions, potentially affecting transformation stability. Addressing these aspects would further enhance the scalability and generality of our framework.

# 7 Acknowledgement

This work was in part supported by the Science and Technology Projects of the Ministry of Agriculture and Rural Affairs of China, the Strategic Priority Research Program of the Chinese Academy of Sciences under Grant No. XDA0450203, and the National Natural Science Foundation of China under Grant 62172392.

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

# A  Technical Appendices and Supplementary Material

This supplementary material provides additional information to support the main text. We present extended methodological details in Section A.1, and elaborate on implementation aspects in Section A.2. Section A.3 reports additional quantitative and qualitative results, while Section A.4 includes further ablation studies.

## A.1  Method Details

**Network architecture.**  Our dynamic MLP model consists of several modular components designed for spatio-temporal deformation prediction and appearance modeling. The input 3D positions and timestamps are first encoded using positional encoders. Specifically, we use 6 frequency bands for both spatial and temporal encodings on the D-NeRF dataset, and 10 frequency bands on the NeRF-DS dataset, in accordance with the scene complexity and temporal resolution. When the Hierarchical Anchor Densification module is enabled, the encoded features are first processed by a lightweight two-layer MLP for anchor-level feature extraction. Notably, for all hierarchical levels beyond the first, the MLP additionally takes the positional encoding of the parent anchor's transformation $\Delta T$ as input. Each level maintains its own independent MLP parameters without weight sharing. The network then processes the extracted features through three parallel branches: motion mask prediction, scene flow estimation, and deformation modeling.

The motion mask branch consists of a compact three-layer MLP that predicts a soft dynamic confidence via a sigmoid activation, which modulates the outputs of subsequent branches during training. The scene flow branch is composed of a four-layer MLP with skip connections and outputs forward and backward displacements to support temporal feature aggregation. The core deformation branch consists of an eight-layer MLP with skip connections, followed by three independent prediction heads that separately regress the translation, scaling, and rotation parameters for each Gaussian.

We assign different numbers of motion anchors to each scene based on its motion complexity. For deformation interpolation, the number of nearest anchors is set to $K=3$. In the Hierarchical Anchor Densification module (Section 4.4), we set the threshold $\tau$ to 0.75 to achieve optimal performance, and adapt the number of hierarchy levels according to the complexity of each scene.

## A.2  More Implementation Details

**Datasets.**  The NeRF-DS dataset comprises seven stereo video sequences capturing everyday scenarios with rapidly moving, reflective objects and varying camera viewpoints, presenting significant challenges for dynamic scene reconstruction. We use the default resolution 480×270 for all scenes for training and testing. We train the model using images from the left camera and test it on the right camera. The D-NeRF dataset comprises eight synthetic dynamic scenes with precise camera trajectories and ground-truth deformations. We use the default resolution 800×800 for all scenes for training and testing. Notably, we exclude the Lego scene from our evaluation due to a mismatch between the training and test sets, particularly in the flip angle of the Lego shovel, which has also been recognized in the Deformable 3DGS [44].

**Optimization.**  The scheduler of the learning rate primarily follows Deformable 3DGS [44] and SC-GS [12]. The entire training process is conducted using the Adam optimizer on a single RTX 3090 GPU. The loss weights in Equation 9 are set to $\lambda = 0.8$, $\lambda_1 = 0.01$, $\lambda_2 = 0.2$, and $\lambda_3 = 0.5$. We further evaluate the effect of different weight settings on reconstruction performance in Section 4.5 using the NeRF-DS dataset, and select the optimal configuration based on empirical results. We use a simple landmark-based interpolation strategy to adjust loss weights over training steps. Given a list of step-value pairs, each weight is computed via logarithmic interpolation between neighboring landmarks, enabling smooth and stage-aware optimization.

## A.3  More Results

We conduct additional quantitative comparisons on the D-NeRF dataset and evaluate all methods using PSNR, SSIM, and LPIPS(VGG) metrics. The results are summarized in Table 4. In addition, we further compare our method with more approaches on the NeRF-DS dataset, and present the quantitative results (PSNR) in Table 5. we conduct a quantitative comparison on additional real-world

Table 4. **Additional quantitative comparison on D-NeRF dataset per-scene.**

| Method | Hook | | | Jumping Jacks | | | Trex | | | Bouncing Balls | | |
|---|---|---|---|---|---|---|---|---|---|---|---|---|
| | PSNR↑ | SSIM↑ | LPIPS↓ | PSNR↑ | SSIM↑ | LPIPS↓ | PSNR↑ | SSIM↑ | LPIPS↓ | PSNR↑ | SSIM↑ | LPIPS↓ |
| TiNeuVox [7] | 30.51 | 0.959 | 0.060 | 33.46 | 0.977 | 0.041 | 31.43 | 0.967 | 0.047 | 40.28 | 0.992 | 0.042 |
| D-NeRF [31] | 29.02 | 0.959 | 0.054 | 32.70 | 0.977 | 0.038 | 30.61 | 0.967 | 0.053 | 38.17 | 0.989 | 0.032 |
| Tensor4D [33] | 28.63 | 0.943 | 0.063 | 24.20 | 0.925 | 0.066 | 23.86 | 0.935 | 0.054 | 24.47 | 0.962 | 0.043 |
| K-Planes [8] | 28.12 | 0.948 | 0.066 | 31.11 | 0.970 | 0.046 | 30.43 | 0.973 | 0.034 | 40.05 | 0.993 | 0.032 |
| 3DGS [14] | 21.70 | 0.886 | 0.104 | 20.64 | 0.929 | 0.106 | 21.91 | 0.953 | 0.055 | 23.20 | 0.958 | 0.060 |
| 4DGS [39] | 32.95 | 0.977 | 0.027 | 35.50 | 0.986 | 0.020 | 33.95 | 0.985 | 0.022 | 40.77 | 0.994 | 0.015 |
| SP-GS [5] | 35.36 | 0.980 | 0.018 | 35.56 | 0.995 | **0.006** | 32.69 | 0.986 | 0.024 | 40.53 | 0.983 | 0.032 |
| Deformable 3DGS [44] | 37.06 | 0.986 | 0.016 | 37.66 | 0.989 | 0.013 | 37.56 | 0.993 | 0.010 | 40.91 | 0.995 | 0.009 |
| SC-GS [12] | 38.79 | 0.990 | **0.011** | 39.34 | 0.992 | 0.008 | 39.53 | 0.994 | 0.009 | 41.59 | 0.995 | 0.009 |
| Ours | **39.38** | **0.996** | 0.013 | **39.79** | **0.997** | 0.010 | **40.27** | **0.998** | **0.009** | **41.63** | **0.996** | **0.009** |

| Method | Hell Warrior | | | Mutant | | | Standup | | | Mean | | |
|---|---|---|---|---|---|---|---|---|---|---|---|---|
| | PSNR↑ | SSIM↑ | LPIPS↓ | PSNR↑ | SSIM↑ | LPIPS↓ | PSNR↑ | SSIM↑ | LPIPS↓ | PSNR↑ | SSIM↑ | LPIPS↓ |
| TiNeuVox [7] | 27.29 | 0.964 | 0.076 | 32.07 | 0.961 | 0.048 | 34.46 | 0.980 | 0.033 | 32.78 | 0.971 | 0.049 |
| D-NeRF [31] | 24.06 | 0.944 | 0.070 | 30.31 | 0.967 | 0.039 | 33.13 | 0.978 | 0.035 | 31.14 | 0.968 | 0.045 |
| Tensor4D [33] | 31.26 | 0.925 | 0.073 | 29.11 | 0.945 | 0.060 | 30.56 | 0.958 | 0.036 | 27.44 | 0.941 | 0.056 |
| K-Planes [8] | 24.58 | 0.952 | 0.082 | 32.50 | 0.971 | 0.036 | 33.10 | 0.979 | 0.031 | 31.41 | 0.969 | 0.046 |
| 3DGS [14] | 29.89 | 0.914 | 0.111 | 24.50 | 0.933 | 0.058 | 21.91 | 0.929 | 0.089 | 23.39 | 0.928 | 0.083 |
| 4DGS [39] | 28.80 | 0.974 | 0.037 | 37.75 | 0.988 | 0.016 | 38.15 | 0.990 | 0.014 | 35.41 | 0.984 | 0.021 |
| SP-GS [5] | 40.19 | 0.989 | **0.006** | 39.43 | 0.986 | 0.016 | 42.07 | 0.992 | 0.009 | 37.97 | 0.987 | 0.015 |
| Deformable 3DGS [44] | 41.34 | 0.987 | 0.024 | 42.47 | 0.995 | 0.005 | 44.14 | 0.995 | 0.007 | 40.16 | 0.991 | 0.012 |
| SC-GS [12] | 42.19 | 0.989 | 0.019 | 43.43 | 0.996 | 0.005 | 46.72 | 0.997 | 0.004 | 41.65 | 0.993 | **0.009** |
| Ours | **42.50** | **0.993** | 0.021 | **43.65** | **0.998** | **0.005** | **46.83** | **0.999** | **0.004** | **42.00** | **0.996** | 0.010 |

Table 5. **Additional quantitative comparison (PSNR↑) on NeRF-DS dataset per-scene.**

| Method | Sieve | Plate | Bell | Press | Cup | As | Basin | Mean |
|---|---|---|---|---|---|---|---|---|
| SP-GS [5] | 25.62 | 18.91 | 25.20 | 24.34 | 24.43 | 24.44 | 19.09 | 23.15 |
| SoM [38] | 26.32 | 20.58 | 25.97 | 25.41 | 24.55 | 26.29 | 19.66 | 24.11 |
| MoSca [17] | 26.45 | **21.13** | 26.11 | 26.53 | **24.91** | 26.17 | **20.28** | 24.51 |
| GauFre [21] | 24.88 | 20.00 | 25.24 | 25.05 | 24.04 | 26.05 | 19.54 | 23.64 |
| MoDec-GS [16] | 23.48 | 18.87 | 22.19 | 22.87 | 24.18 | 24.65 | 19.57 | 22.25 |
| **Ours** | **26.61** | 20.96 | **26.34** | **27.05** | 24.72 | **26.96** | 19.74 | **24.63** |

datasets: HyperNeRF(misc) [30] and Dycheck [10], and the PSNR results are presented in Table 6 and Table 7.

We report a comparison of average training computational cost between our method and existing approaches on the D-NeRF dataset, including training time, rendering FPS, and memory in Table 8, as well as detailed per-scene results in Table 9. We also report the statistics on NeRF-DS dataset in Table 10.

The additional qualitative results on the NeRF-DS dataset are provided in Figure 6. The additional qualitative results on the D-NeRF dataset are provided in Figure 7.

## A.4 More Ablations

We summarize the additional ablation results of different combinations of key components in Table 11. In main paper, we have already provided ablation studies on different densification strategies. Figure 5 presents qualitative results on regions with fine-grained motion, demonstrating that the HAD module effectively improves reconstruction quality in these areas. As shown in Table 11, extending the Baseline+IF setup with the HAD module (row 4) results in a noticeable performance gain, further supporting the effectiveness of HAD. And as shown in Table 11, extending the Baseline+IF setup with the AF module (row5) also leads to a performance improvement, indicating that AF effectively guides the Induced Flow MLP and Deformable MLP to focus on dynamic regions, thereby enhancing overall model performance. These ablations validate the design motivation of HAIF-GS.

We conduct additional ablation studies to evaluate the impact of the three loss terms introduced in our framework. The results are summarized in Table 12. It should be emphasized that $\lambda = 0$ denotes the exclusion of the respective loss term.

Table 6. **Additional quantitative comparison (PSNR↑) on HyperNeRF(misc) dataset per-scene.**

| Method | Americano | Cross-hands | Espresso | Keyboard | Oven-mitts | Split-cookie | Tamping | Mean |
|---|---|---|---|---|---|---|---|---|
| 4DGS [39] | **31.30** | 28.06 | 25.82 | **28.64** | **27.99** | 32.64 | **24.15** | 28.37 |
| Deformable 3DGS [44] | 30.87 | 27.70 | 25.47 | 28.15 | 27.51 | 32.63 | 23.95 | 28.04 |
| **Ours** | 30.94 | **28.83** | **26.58** | 28.54 | 27.65 | **33.20** | 23.79 | **28.50** |

Table 7. **Additional quantitative comparison (PSNR↑) on Dycheck dataset per-scene.**

| Method | Apple | Block | Paper-windmill | Space-out | Spin | Teddy | Wheel | Mean |
|---|---|---|---|---|---|---|---|---|
| 4DGS [39] | 15.41 | 13.89 | 14.44 | 14.29 | 14.89 | 12.31 | 10.53 | 13.68 |
| Deformable 3DGS [44] | **15.61** | **14.87** | 14.89 | 14.59 | 13.10 | 11.20 | 11.79 | 13.72 |
| **Ours** | 15.51 | 14.64 | **16.13** | **15.45** | **16.59** | **12.91** | **14.12** | **15.05** |

Table 8. **Overall performance comparison on the D-NeRF dataset**, including training time, rendering FPS, and GPU memory.

| Method | PSNR | Training Time | FPS | Memory |
|---|---|---|---|---|
| 4DGS [39] | 35.41 | 26 min | 134.8 | 1.1 GB |
| Deformable 3DGS [44] | 40.16 | 38 min | 53.7 | 4.8 GB |
| **Ours** | **42.00** | 29 min | **215.4** | 3.3 GB |

Table 9. **Per-scene performance on the D-NeRF dataset.**

| Scene | PSNR↑ | Queried Anchors | Training Time↓ | FPS↑ |
|---|---|---|---|---|
| Hook | 39.38 | 604 | 38 min | 211.4 |
| Jumpingjacks | 39.79 | 560 | 26 min | 259.8 |
| Trex | 40.27 | 792 | 41 min | 165.2 |
| BouncingBalls | 41.63 | 524 | 21 min | 176.3 |
| Hellwarrior | 42.50 | 568 | 23 min | 236.5 |
| Mutant | 43.65 | 588 | 35 min | 208.8 |
| Standup | 46.83 | 572 | 27 min | 249.5 |
| Mean | 42.00 | 601 | 29 min | 215.4 |

Table 10. **Statistics of Motion Anchors on NeRF-DS.**

| Scene | Sieve | Plate | Bell | Press | Cup | As | Basin | Mean |
|---|---|---|---|---|---|---|---|---|
| #Gaussians | 114k | 119k | 185k | 114k | 118k | 113k | 130k | 127k |
| #Anchors | 3.7k | 4.3k | 4.4k | 3.5k | 3.9k | 4.1k | 4.9k | 4.1k |
| Ratio | 0.287 | 0.314 | 0.246 | 0.253 | 0.279 | 0.304 | 0.325 | 0.287 |

Table 11. **Additional ablation study of key component combinations on NeRF-DS dataset.**

| Method | PSNR↑ | MS-SSIM↑ | LPIPS↓ |
|---|---|---|---|
| **Baseline** | 23.79 | 0.8513 | 0.1744 |
| + IF (w/o $L_{cycle}$) | 24.15 | 0.8705 | 0.1561 |
| + IF | 24.28 | 0.8721 | 0.1505 |
| + IF + HAD | 24.45 | 0.8802 | 0.1488 |
| + IF + AF | 24.39 | 0.8774 | 0.1493 |
| + IF (w/o $L_{cycle}$) + AF + HAD | 24.51 | 0.8947 | 0.1437 |
| **Ours** (+IF + AF + HAD) | **24.63** | **0.9014** | **0.1342** |

Table 12. **Additional ablation study of the loss weights on NeRF-DS dataset.**

| $\lambda_{\text{cycle}}$ | $10^1$ | $10^{-1}$ | $10^{-2}$ | $10^{-3}$ | $10^{-4}$ | 0 |
|---|---|---|---|---|---|---|
| PSNR↑ | 24.53 | 24.58 | **24.63** | 24.60 | 24.59 | 24.51 |

| $\lambda_{\text{entropy}}$ | $10^1$ | $10^0$ | $10^{-1}$ | $10^{-2}$ | $10^{-3}$ | 0 |
|---|---|---|---|---|---|---|
| PSNR↑ | 24.58 | 25.57 | **24.62** | 24.60 | 24.58 | 24.55 |

| $\lambda_{\text{sparsity}}$ | $10^1$ | $10^0$ | $10^{-1}$ | $10^{-2}$ | $10^{-3}$ | 0 |
|---|---|---|---|---|---|---|
| PSNR↑ | 24.60 | 24.63 | **24.63** | 24.58 | 24.58 | 24.56 |

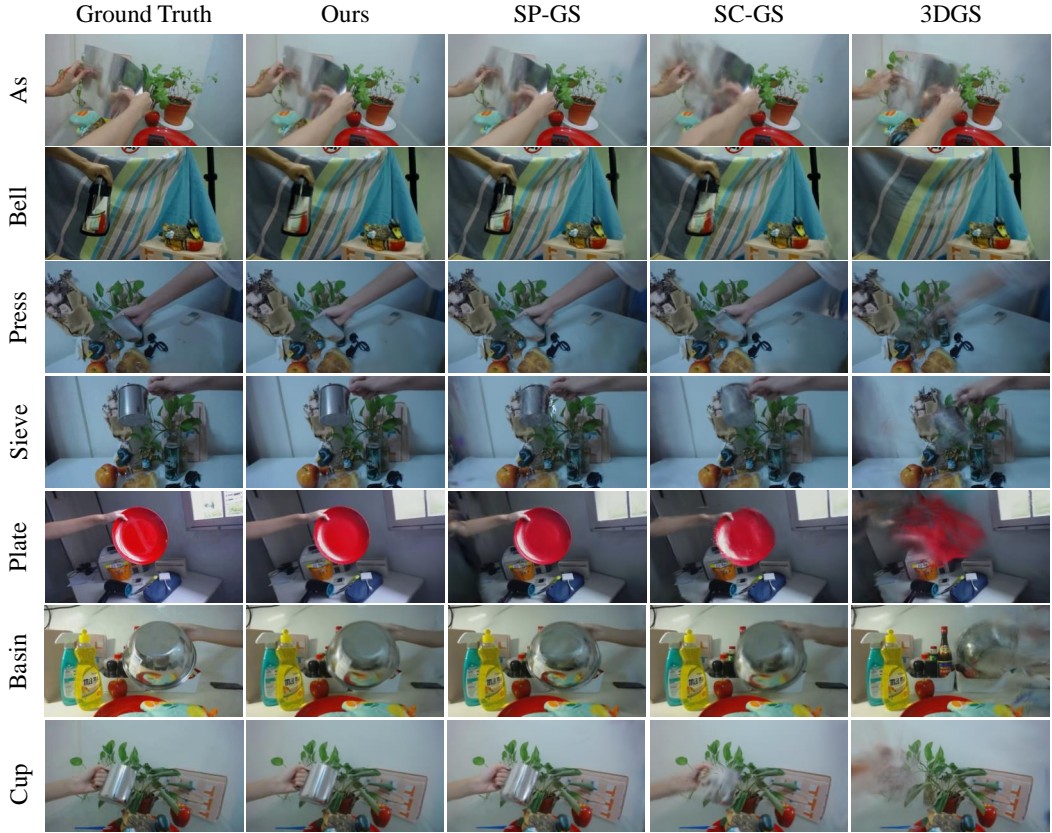

Figure 6. **Additional qualitative comparison on the NeRF-DS dataset [42].**

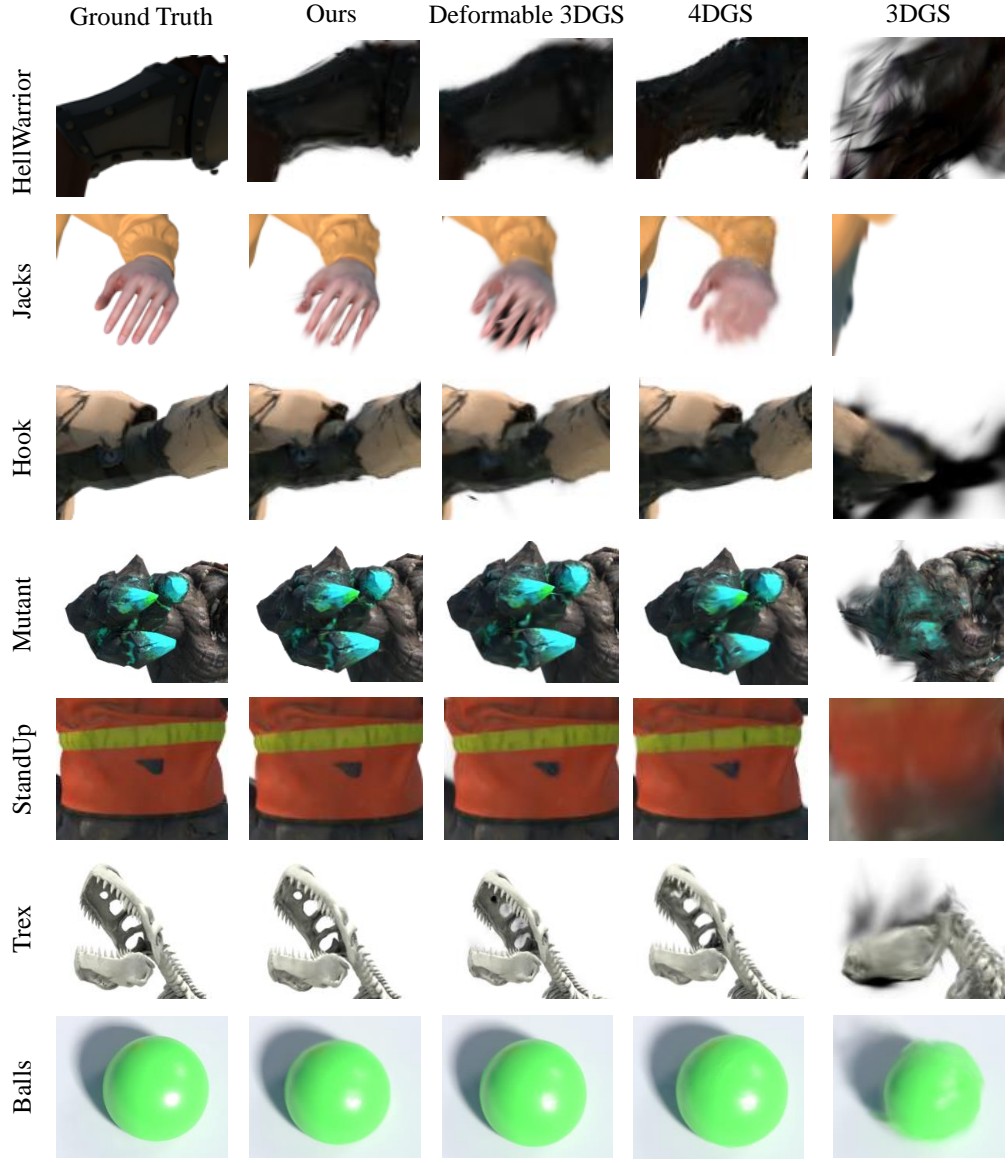

Figure 7. **Additional qualitative comparison on the D-NeRF dataset [31]**.

