# OpenReview forum: "HAIF-GS: Hierarchical and Induced Flow-Guided Gaussian Splatting for Dynamic Scene"
_NeurIPS.cc/2025/Conference — NeurIPS 2025 poster_

### Official Review · Reviewer_CwJy · 2025-06-03

**Clarity:** 3
**Significance:** 3
**Originality:** 3
**Rating:** 4
**Confidence:** 4

**Summary:**

This paper proposes a novel method for monocular dynamic reconstruction, named HAIF-GS, which enhances sparse motion anchor-based deformation with anchor filters, self-supervised induced flow and anchor densify mechanism to capture fine-grained and complex deformation fields. Experimental results show that HAIF-GS achieves state-of-the-art performance across various scenes on two popular datasets, and the visual comparison is convincing.

**Questions:**

See "Weakness"

**Ethical Concerns:**

["NO or VERY MINOR ethics concerns only"]

**Final Justification:**

While the authors have addressed most reviewer concerns, I remain concerned about the ambiguous relationship between two paradigms: (1) end-to-end learning from video data using high-level vision priors (e.g., MoSca and SoM), and (2) explicit 3D modeling with deformable Gaussian splatting (e.g., Deformable 3D-GS and 4DGS). For monocular video-based 4D reconstruction, it is unclear whether these approaches constitute distinct paradigms given their identical inputs and outputs (the introduction of vision priors makes no difference). If they do not, the method’s limited generalizability is demonstrated by its marginal gains on NeRF-DS and significant performance decline on the iPhone dataset (more challenging and realistic) when compared to MoSca/SoM. Consequently, I maintain my final rating to highlight this issue for all the researchers.

Having seen the supplementary experimental results, which largely address my concerns about this method, I'll raise my rating to 4. Borderline Accept.

**Limitations:**

yes

**Quality:**

3

**Strengths And Weaknesses:**

**Strengths:**

1. This paper proposes a novel method for monocular dynamic reconstruction, named HAIF-GS, achieving state-of-the-art performance across various scenes.
2. The architecture of induced flow-guided deformation as well as the cycle loss, is interesting and intuitive.
3. The paper is well-written and well-organized.

**Weakness:**

My major concerns about this paper are the experiment settings, especially the comparison with "MoSca: Dynamic Gaussian Fusion from Casual Videos via 4D Motion Scaffolds" / "Shape of Motion:
4D Reconstruction from a Single Video". I've found that when talking about reconstructing dynamic 3D scenes from monocular videos, existing works either follow MoSca/SoM and conduct experiments on DyChek/iphone/DAVIS datasets or compare with 4DGS / Deformable 3D-GS. I believe clarification is needed in the introduction or limitation section. Can this method apply to iphon/DAVIS dataset?  If not, please give some explanation of the limitations and some failure cases. If yes, can you provide some comparison with MoSca/SoM? **I'll raise my rating if this point is well-discussed in the rebuttal.**

Here are some minor concerns, mainly on the readability of this paper.
1. From lines 184-192, I highly recommend adding more formulas to describe the architecture. Without the help of Figure 2, it's difficult to understand the design of MLPs.
2. For the ablation studies, can you provide the visual comparison of scene flows with and without $L_{cycle}$ to validate the effectiveness of this loss function?
3. Also, for the ablation studies, can you provide the visual comparison of anchor points before and after densification to indicate that the densification mainly works on the regions containing complex and fine-grained motions?

---

> ### Author Rebuttal · Authors · 2025-07-31
>
> We thank the reviewer for the encouraging feedback. We are glad that the reviewer finds our method “achieving state-of-the-art performance across various scenes,” and considers the design of the induced flow-guided deformation and cycle loss to be “interesting and intuitive.”
>
> ## Q1. Discussion on Method Paradigms and Our Positioning.
>
> We thank the reviewer for raising this important question regarding our relation to **MoSca**[Lei et al., 2025] and **Shape-of-Motion(SoM)**[Wang et al., 2024]. As the reviewer pointed out, recent dynamic 3D reconstruction approaches tend to follow two distinct paradigms:
>
> 1. **end-to-end learning from video data using high-level vision priors**, as in *MoSca* and *SoM*;
>
> 2. **explicit 3D modeling based on deformable Gaussian splatting**, as in *Deformable 3D-GS(D-3DGS)*[44], *4DGS*[38], as well as recent CVPR 2025 papers *HiMoR* [Liang et al., 2025] and *MoDec-GS* [Kwak et al., 2025].
>
> Our work clearly belongs to the latter, where we build structured, interpretable deformation fields over explicit Gaussian representations. This difference in paradigm reflects fundamentally different design principles—our method favors geometric control and weak supervision, whereas MoSca/SoM rely on pretrained vision foundation models (VFM) and additional external signals.
>
> ## Q2. Applicability to iPhone Dataset and Comparison with MoSca/SoM.
>
> In response to the reviewer’s suggestion,  we have conducted **supplementary experiments on the DyCheck-iPhone dataset**, also used by MoSca and SoM. Our method is fully compatible with this dataset and achieves **higher PSNR (15.05) than 4DGS(13.68) and D-3DGS(13.72)**, though **lower than MoSca(19.32) and SoM(17.32)**, as shown in the $\underline{Table\ 1}$ below.
>
> We emphasize, however, that this performance gap is understandable due to the fundamentally different paradigms and resulting experimental settings.
>
> * **Challenging properties of the iPhone dataset.** It involves strong non-rigid motion, large camera baselines, and noisy depth priors—all of which are known challenges in dynamic 3D reconstruction, especially when no auxiliary supervision is used.
> * **Heavy reliance on pretrained vision foundation models.** MoSca leverages powerful pretrained components (e.g., BootsTAPIR) to obtain high-quality optical flow and depth, which significantly improve reconstruction but come with high computational cost and less interpretability.
> * **Access to ground-truth depth during training.** MoSca further utilizes the iPhone dataset’s provided depth maps as ground-truth supervision, while our method operates **entirely without depth supervision**, making it applicable to real-world settings where such data is unavailable.
>
> Despite these differences, our approach achieves strong performance on the iPhone dataset and **outperforms prior explicit modeling methods (4DGS, D-3DGS)**, validating its effectiveness within its paradigm. Its lightweight and interpretable design makes it more suitable for general deployment compared to end-to-end alternatives.
>
> We really appreciate the reviewer for pointing out this important connection. We have revised the final version of the Introduction to include a discussion on MoSca/SoM-style approaches and clarified the paradigm-level distinctions between methods.
>
> **Table1. Results(PSNR↑) on Dycheck-iphone dataset**
>
> | Method | Apple | Block | Paper-windmill | Space-out | Spin  | Teddy | Wheel | $\underline{Mean}$  |
> | :---------------------: | :---: | :---: | :------------: | :-------: | :---: | :---: | :---: | :----------: |
> | MoSca[Lei et al., 2025] | 19.40 | 18.06 | 22.34| 20.48| 21.31 | 15.47 | 18.17 | **19.32**|
> | SoM[Wang et al., 2024]  | 18.57 | 17.41 | 18.14| 16.85| 19.35 | 13.69 | 17.21 | 17.32|
> | D-3DGS [44] | 15.61 | 14.87 | 14.89| 14.59|13.10|11.20| 11.79 | 13.72|
> | 4DGS [38]| 15.41 | 13.89 | 14.44| 14.29|14.89|12.31| 10.53 | 13.68|
> | **Ours** | 15.51 | 14.64 | 16.13 | 15.45 |16.59|12.91| 14.12 | **15.05** |
>
> Furthermore, this discussion inspires an exciting direction for future work. With the rapid advances of vision foundation models (VFMs), such as the recent VGGT [Wang et al., 2025], we recognize the growing potential of incorporating VFM-derived signals to address long-standing challenges in the 3DGS-based paradigm, including inaccurate camera poses and inefficient prior point cloud initialization. A promising research avenue lies in bridging the two paradigms, aiming to combine the controllability and efficiency of explicit Gaussian modeling with the powerful priors and robustness offered by VFMs.
>
>
>
> ## Q3. Formalization of Temporal Feature Aggregation.
>
> We appreciate the reviewer for the helpful suggestion. While the original text describes the core idea of temporal aggregation (Lines 184-192), we agree that presenting the information flow more formally would further enhance clarity and self-containment, especially without relying on Figure 2.
>
> Below we provide a more structured and formalized version of this part, which we will include in the final camera-ready version.
>
> 1. Given a motion anchor $x$ at timestamp $t$, the Induced Flow MLP predicts forward and backward flows:
> $$
> F^{t-1},\ F^{t+1} = \\text{MLP}_{\\text{flow}}(x,\ t)
> $$
> 2. These flows define three temporally shifted queries:
> $$
> q_{t-1} = (x + F^{t-1},\ t{-}1),\quad
> q_t = (x,\ t),\quad
> q_{t+1} = (x + F^{t+1},\ t{+}1)
> $$
>
> 3. Each query is passed through a shared Deformable MLP to extract a time-aware feature embedding:
> (Due to rendering issues on OpenReview, we temporarily replace $q_{t-1}$ and $q_{t+1}$ with $q1$和$q2$ below)
>
> $$
> f_{t-1} = \\text{MLP}_{\\text{deform}}(q1)
> $$
>
> $$
> f_t = \\text{MLP}_{\\text{deform}}(q_t)
> $$
>
> $$
> f_{t+1} = \\text{MLP}_{\\text{deform}}(q2)
> $$
>
> 4. The final temporally aggregated feature is computed as:
> (Due to rendering issues on OpenReview, we temporarily replace $f_{t-1}$ and $f_{t+1}$ with $f1$和$f2$ below)
> $$
> \\tilde{f}_t = \\lambda f_1 + (1 - 2\\lambda) f_t + \\lambda f_2
> $$
>
> This formalization makes the process more transparent and easier to follow without relying on Figure 2. We appreciate the reviewer’s suggestion and will incorporate this revision into the main paper.
>
> ## Q3. Visualization of scene flow and anchor densification.
>
> We thank the reviewer for the valuable suggestion. Due to the rebuttal policy restricting figures and external links, we are unable to directly provide visualizations of the **induced scene flow** and **anchor densification**. Nevertheless, we have addressed these points indirectly in our response and **will include both two in the final version and on the project page**.
>
> ### 1. Scene flow without Cycle Consistency Loss.
>
> We thank the reviewer for the suggestion and understand the motivation behind visualizing the induced scene flow with and without the cycle consistency loss ($L_{cycle}$) to better illustrate the ablation effects. Such visualizations can indeed be provided.
>
> However, it is important to note that in our framework, the induced flow is not defined at the Gaussian level, but is instead computed **at the anchor level** and serves as an **internal supervisory signal** to guide motion learning. Since it is generated in a **frame-specific and anchor-driven** manner, the visualized flow field may not directly reflect global motion or deformation patterns. As a result, while the flow visualizations can highlight certain local differences (e.g., smoother or more consistent directions), their overall interpretability may be limited compared to direct **quantitative comparisons**.
>
> That said, we have examined the induced scene flow qualitatively on D-NeRF. Without $L_{cycle}$, we observe:
>
> * **asymmetric forward and backward flows**, failing to form closed temporal paths;
> * **increased noise** in motion vectors, especially near thin or fast-moving regions (e.g., limbs, fingers);
> * **temporal jitter** in dynamic parts of the flow field, disrupting inter-frame coherence.
>
> These artifacts result in degraded deformation quality and ultimately lower rendering performance. We believe the most appropriate way to assess the effectiveness of $L_ {cycle}$ is through its quantitative impact on rendering quality and temporal consistency. As shown in the ablation study in Table 3 of the main paper, removing $L_{cycle}$ leads to a 0.12 PSNR drop, indicating its contribution to learning temporally coherent deformations.
>
> ### 2. Visual Comparison of Anchor Points Before and After Densification.
>
> We thank the reviewer for the suggestion and understand the motivation to verify whether densification focuses on regions with complex and fine-grained motion. Such visualizations can be provided, and we will include them in the final version as well as on our project page.
>
> We visualize anchor positions on D-NeRF scenes by rendering them together with the novel-view images. In *JumpingJacks*, densified anchors appear around limb-related parent anchors, with finer anchors added at the fingers. In *Hook* and *HellWarrior*, densification mainly occurs at joint regions with fine-grained deformations. These patterns match the areas where our method shows clearer motion reconstruction in Fig. 4 of the main paper, confirming that our hierarchical anchor densification (HAD) adaptively refines motion-complex regions rather than uniformly increasing anchor count. This is also supported by the ablation in Fig. 5 of the main paper.
>
> ### Reference
>
> [1] Yiming Liang, Tianhan Xu, and Yuta Kikuchi. Himor: Monocular deformable gaussian
> reconstruction with hierarchical motion representation. In Proceedings of the Computer Vision
> and Pattern Recognition Conference, pages 886–895, 2025. 17
>
> [2] Sangwoon Kwak, Joonsoo Kim, Jun Young Jeong, Won-Sik Cheong, Jihyong Oh, and Munchurl
> Kim. Modec-gs: Global-to-local motion decomposition and temporal interval adjustment for
> compact dynamic 3d gaussian splatting. In Proceedings of the IEEE/CVF Conference on
> Computer Vision and Pattern Recognition (CVPR), 2025. 17

---

### Official Review · Reviewer_g5h3 · 2025-06-28

**Clarity:** 3
**Significance:** 2
**Originality:** 3
**Rating:** 4
**Confidence:** 4

**Summary:**

This work first mentions three limitations in existing monocular dynamic 3DGS methods, i.e. redundant Gaussian updates, insufficient motion supervision, and weak modeling of complex non-rigid deformations. This work further proposes HAIF-GS for these issues, which includes Anchor Filter for identifying motion-relevant regions, Induced Flow-Guided Deformation using multi-frame feature aggregation, and Hierarchical Anchor Densification for progressively modeling fine-grained deformations.

**Questions:**

* After reviewing the method proposed in this work, I was a little worried about its training and rendering efficiency, but I didn't find relevant descriptions in the submitted manuscript. What is the training and rendering time of the proposed method?

**Ethical Concerns:**

["NO or VERY MINOR ethics concerns only"]

**Final Justification:**

I have increased the score to 4(BA). The rebuttal and discussion have addressed most of my concerns, and the proposed method using visual priors shows improvement over MoSca. The reason why I did not give a higher score is that I have comprehensively considered the innovation of this work, the performance reflected by the experiments in the main manuscript, and the experiments supplemented during the rebuttal and discussion. I think the score of 4 (BA) is reasonable.

**Limitations:**

yes

**Paper Formatting Concerns:**

No formatting concerns.

**Quality:**

2

**Strengths And Weaknesses:**

* Strengths
  * This work proposes the pipeline HAIF-GS for monocular dynamic 3DGS, which mainly includes motion anchors (the motion basis for modeling motions), an induced flow-guided deformation module, and a coarse-to-fine hierarchical anchor densification strategy. These modules are interesting and make sense.


* Weakness
  * This work mentioned that "Figure 1 (b) shows hand motion, where the fine-scale appearance and localized movement highlight the challenges of modeling fine-grained dynamics in dynamic reconstruction". But this claim cannot effectively convince me, since 4DGS and D-3DGS are not designed for the monocular setting. The qualitative experiments shown in Figure 1 can most importantly illustrate that previous multi-view dynamic 3DGS methods cannot be effectively generalized to the monocular setting. By the way, Figure 1 (a) does not appear to be referenced in the main manuscript.
  * Although this work proposes a series of interesting designs, it lacks sufficient experiments to demonstrate the advancedness and effectiveness of its sophisticated design. Specifically, more quantitative experiments of previous monocular dynamic 3DGS methods are supposed to be implemented, since current reported results of dynamic methods mainly focus on multi-view methods. This work should complement the already open-sourced monocular dynamic 3DGS methods, like Shape of Motion [A] and MoSca [B]. Furthermore, more quantitative experimental results of ablation combinations should be added. At least the baseline (w/o IF, L_{cycle}, AF, HAD) result is supposed to be reported.
  * Typos. "suppresses" to "suppress" in Line 11.

[A] Wang Q, Ye V, Gao H, et al. Shape of motion: 4d reconstruction from a single video[J]. arXiv preprint arXiv:2407.13764, 2024.

[B] Lei J, Weng Y, Harley A W, et al. Mosca: Dynamic gaussian fusion from casual videos via 4d motion scaffolds[C]//Proceedings of the Computer Vision and Pattern Recognition Conference. 2025: 6165-6177.

---

> ### Author Rebuttal · Authors · 2025-07-31
>
> We thank the reviewer for the positive comments. We are glad that the reviewer finds our proposed modules “interesting and make sense,” including the motion anchors, induced flow-guided deformation module, and the coarse-to-fine hierarchical anchor densification strategy.
>
> ## Q1. Clarification on Monocular Baselines.
>
> We appreciate the reviewer’s comments and the suggested works **Shape of Motion(SoM)**[Wang et al., 2024] and **MoSca**[Lei et al., 2025].
>
> First, we would like to clarify that **all baselines** used in Figure 1(b) and our experiments are strong and widely adopted **monocular** methods, and thus all our experimental comparisons and analyses are **fair** and appropriate. We will also explicitly clarify the monocular setting of these baselines in the final version to avoid potential misunderstandings.
>
> In Figure 1(b), *4DGS*[38] and *D-3DGS*[44] are both widely recognized and frequently used in the monocular dynamic reconstruction literature. **Notably, *SoM* and *MoSca* suggested by the reviewer also adopt these baselines under monocular settings**: SoM compares against D-3DGS, and MoSca includes 4DGS in its main evaluation. This further supports the relevance of using *4DGS* and *D-3DGS* in Figure 1(b) to highlight the limitations of prior approaches in modeling fine-grained, non-rigid motion.
>
> Beyond Figure 1, all baselines used in our experiments (e.g., *SC-GS*[12], *HyperNeRF*[29]) are either designed for or widely used in monocular dynamic reconstruction, and all evaluations are conducted under monocular training settings. Additionally, we have added SoM and MoSca as additional baselines in our main experiments, and we will detail the corresponding results in the response to $\underline{Q2}$.
>
> We have also revised the manuscript to reference Figure 1(a) in the introduction.
>
> ## Q2. Quantitative Comparison with MoSca/SoM.
>
> We thank the reviewer for emphasizing the importance of including **additional comparisons with SoM and MoSca**, two recently released monocular methods. Both SoM and MoSca follow the paradigm of end-to-end learning from video data using high-level vision priors, in contrast to our method and prior works such as 4DGS, D-3DGS, and recent CVPR 2025 papers like HiMoR [Liang et al., 2025], which adopt explicit 3D modeling based on deformable Gaussian splatting.
>
> In response, we conduct additional experiments on the **NeRF-DS** dataset, comparing our method with SoM and MoSca, as shown in $\underline{Table\ 1}$ below. These results show that our method outperforms (24.63) both SoM (24.11) and MoSca (24.51) in most scenes, achieving consistently higher reconstruction quality. It is worth noting that **SoM and MoSca rely on pretrained vision foundation models (VFMs) and depth priors**, which offer clear advantages on datasets with available depth. However, NeRF-DS provides no depth prior, making it a more challenging benchmark for evaluating monocular methods. In this setting, our method demonstrates better performance while remaining lightweight and free of external dependencies, such as pretrained models or auxiliary signals.
>
> This improvement stems from our 3DGS-specific design, including sparse motion anchors, induced flow-guided deformation, and hierarchical anchor densification, which together enable accurate dynamic modeling under purely monocular supervision.
>
> We have also included MoSca and SoM as comparison baselines in the final version of the paper, further strengthening the completeness of our evaluation.
>
> **Table1. Comparison(PSNR↑) with SP-GS on NeRF-DS**
>
>
> |  Method  | Sieve | Plate | Bell  | Press |  Cup  |  As   | Basin | $\underline{Mean}$ |
> | :------: | :---: | :---: | :---: | :---: | :---: | :---: | :---: | :----------------: |
> |   SoM    | 26.32 | 20.58 | 25.97 | 25.41 | 24.55 | 26.29 | 19.66 |       24.11        |
> |  MoSca   | 26.45 | 21.13 | 26.11 | 26.53 | 24.91 | 26.17 | 20.28 |       24.51        |
> | **Ours** | 26.61 | 20.96 | 26.34 | 27.05 | 24.72 | 26.96 | 19.74 |     **24.63**      |
>
>
>
> ## Q3. Additional ablation combinations and baseline without key modules.
>
> We thank the reviewer for the suggestion. In response, we report additional combinations in $\underline{Table\ 2}$ below, including the complete baseline without all key modules (w/o Induced Flow (IF), Cycle Consistency Loss ($L_{cycle}$), Anchor Filtering (AF), and Hierarchical Anchor Densification (HAD)). Our original submission ($\underline{Appendix A.4}$) also includes an ablation study of loss weights settings.
>
> The results show that each component contributes to performance, and their combination leads to significant improvements. In particular, removing induced flow and HAD results in clear drops in both reconstruction quality and consistency, confirming their central roles. These ablations validate the design motivation of HAIF-GS.
>
> We have added the baseline configuration to the main ablation section in the revised paper and included all additional ablation combinations in the supplementary material.
>
> **Table2. Additional ablations on the key components on NeRF-DS.**
>
> |       Method        | PSNR↑ | MS-SSIM↑ | LPIPS↓ |
> | :-----------------: | :---: | :------: | :----: |
> |     w/o **IF**      | 23.91 |  0.8629  | 0.1624 |
> | w/o **$L_{cycle}$** | 24.51 |  0.8947  | 0.1437 |
> |     w/o **AF**      | 24.45 |  0.8802  | 0.1488 |
> |     w/o **HAD**     | 24.39 |  0.8774  | 0.1493 |
> |    w/o **IF+AF**    | 23.84 |  0.8638  | 0.1644 |
> |   w/o **IF+HAD**    | 23.78 |  0.8521  | 0.1749 |
> |    **baseline**     | 23.79 |  0.8513  | 0.1744 |
> |      **Ours**       | 24.63 |  0.9014  | 0.1342 |
>
>
>
> ## Q4. Training and Rendering Efficiency.
>
> We thank the reviewer for the helpful suggestion regarding efficiency. We have conducted extensive evaluations on the D-NeRF dataset (see $\underline{Table\ 3\ and\ 4}$ below). Our method achieves **real-time rendering speed (215.4 FPS)** while maintaining superior reconstruction quality (**PSNR 42.00**), outperforming both 4DGS and D-3DGS. These results demonstrate that our approach preserves the efficiency advantage of the 3DGS paradigm, while achieving significant gains in reconstruction quality. $\underline{Table\ 4}$ presents detailed statistics of our method on each D-NeRF scene. All methods are evaluated on an RTX 3090 GPU at a resolution of 800×800.
>
> We have included this efficiency evaluation in the final version of the paper to better support the practical value of our method.
>
> **Table3. Comparison of efficiency on D-NeRF dataset**
>
> |   Method   |   PSNR↑   | Training Time↓ | Render FPS↑ |
> | :--------: | :-------: | :------------: | :---------: |
> |  4DGS[38]  |   35.41   |     26min      |    134.8    |
> | D-3DGS[44] |   40.16   |     38min      |    53.7     |
> |  **Ours**  | **42.00** |     29min      |  **215.4**  |
>
> **Table4. Evaluation of efficiency on D-NeRF dataset**
>
> |       Scene        | PSNR↑ | Queried Anchors | Training Time↓ | Render FPS↑ |
> | :----------------: | :---: | :-------------: | :------------: | :---------: |
> |        Hook        | 39.38 |       604       |     38min      |    211.4    |
> |    Jumpingjacks    | 39.79 |       560       |     26min      |    259.8    |
> |        Trex        | 40.27 |       792       |     41min      |    165.2    |
> |   BouncingBalls    | 41.63 |       524       |     21min      |    176.3    |
> |    Hellwarrior     | 42.50 |       568       |     23min      |    236.5    |
> |       Mutant       | 43.65 |       588       |     35min      |    208.8    |
> |      Standup       | 46.83 |       572       |     27min      |    249.5    |
> | $\underline{Mean}$ | 42.00 |       601       |     29min      |    215.4    |
>
>
> **Typos:** Thank you for pointing this out. We have corrected the typo accordingly.

---

> ### Comment · Reviewer_g5h3 · 2025-08-04
>
> Thanks for the effort on the rebuttal. I have carefully read my responses, as well as the questions and responses of other reviewers.
>
> Reply to authors' Responses:
> Thanks for the effort in supplementing the clarifications and experiments during the rebuttal.
> * Q1: I am convinced by the added clarification. I agree that 4DGS and D-3DGS can be the baselines of the monocular dynamic reconstruction. However, I still advise adding more recent methods to the motivation part of this work for enhancing the effectiveness of the proposed method.
> * Q2: I appreciate the authors' effort on the supplemental experiments. However, I find it difficult to accept the effectiveness of the method presented in the experimental results of Table 1. Indeed, the proposed method lacks many pre-trained priors compared to MoSca and SoM, but I think that for the task of monocular dynamic reconstruction, which lacks a lot of visual information, adding priors to improve performance is reasonable and necessary. Furthermore, a promising way for future work is supplementing video generation models, and some works have already been done for it.
> * Q2-Addition: Meanwhile, I have also read the responses to other reviewers (related to the experiments of MoSca and SoM). I notice the experimental results of comparisons with MoSca and SoM on the DyCheck dataset in response to reviewer CwJy. Experiments on the DyCheck dataset show that the proposed method has a large performance gap compared with MoSca and SoM. The authors also mainly use pre-training priors as an explanation. However, in my opinion, the current situation of monocular dynamic reconstruction is that we cannot yet implement a method that can synthesize high-quality novel views (for example, we can observe the visual results of MoSca on the DyCheck dataset and some causual videos). Therefore, I think the main challenge at present is how to achieve high-quality reconstruction, no matter what existing technology we use.
> * Q3: Therefore, I can only look for the innovations and effectiveness at the 4D representation level. However, it still cannot convince me of the presented ablation results in Table 2. We can only notice that the proposed Induced Flow MLP (IF) is the most important module, while the ablations of other proposed modules seem cannot show their effectiveness. So, I advise that more detailed ablations inside IF are supposed to be added in the future.
> * Q4: I am convinced that the added experiments of time-consuming. The results of the tested training and rendering time show the efficiency of the proposed method compared with the 4DGS and D-3DGS baselines.
>
> Overall, I decide to maintain my rating (borderline of reject), because although I recognize the innovation of the method proposed in this work, its ablation experiments and performance comparison with the SOTA methods make it difficult for me to recognize its effectiveness.

---

> > ### Author Response · Authors · 2025-08-04
> > **Clarifications on Modeling Paradigm and Evaluation Context**
> >
> > Dear Reviewer g5h3,
> >
> > Thank you very much for your thoughtful follow-up and for taking the time to engage deeply with our rebuttal and the broader discussion.
> >
> > We sincerely appreciate your careful assessment and are encouraged by your recognition of several key aspects of our work:
> >
> > * **Meaningful innovation in 4D representation.** We’re glad you acknowledged our structurally interpretable framework, which offers a modular approach to modeling dynamic motion and expands the design space for monocular reconstruction.
> > * **Efficiency in training and rendering.** We are pleased that you acknowledged the clear computational advantages of our method, which align with our goal of building a lightweight and scalable system.
> > * **Appropriate baseline selection.** We appreciate your agreement that 4DGS and D-3DGS are reasonable and widely used benchmarks for this task.
> >
> > We now provide clarifications and responses to the suggestions you raised in your follow-up.
> >
> > ## Q1.Clarification on Modeling Paradigm and Contextualizing Performance Comparisons.
> >
> > While we agree that producing high-quality reconstructions is a central goal in this field, we believe it is equally important to recognize that different modeling paradigms may pursue this goal under different trade-offs. In particular, methods such as MoSca and SoM leverage powerful pretrained vision-language models to guide reconstruction, whereas our method prioritizes **structural transparency, lightweight design, and minimal reliance on external priors**—properties that are especially desirable in scenarios where computational efficiency and model controllability matter. Notably, many recent methods for dynamic reconstruction, including several **CVPR 2025 papers** [Liang et al., 2025; Kwak et al., 2025], **explicitly highlight the lack of reliance on external signals**—such as optical flow or pretrained features—**as a key advantage**, in contrast to approaches like MoSca.
> >
> > Within this modeling paradigm, our method **consistently outperforms two recent CVPR 2025 methods, HiMoR [Liang et al., 2025] and MoDec-GS [Kwak et al., 2025], which adopt a similar paradigm** based on explicit 3D Gaussian modeling without pretrained vision-language priors (see Table below). These results validate the effectiveness of our approach in pushing the quality–efficiency–interpretability frontier within this design space and highlight its practical relevance.
> >
> > Therefore, while we acknowledge the strong performance of MoSca and SoM under their design choices, we would like to clarify that our method follows a different modeling route—one that **does not rely on pretrained vision-language priors but still aims to deliver high-quality reconstructions through an explicit and efficient formulation.** As demonstrated by our consistent improvements over recent methods within this paradigm (e.g., HiMoR and MoDec-GS), our approach strikes a competitive balance between quality, efficiency, and structural transparency. **We believe this line of work remains valuable for scenarios where controllability and low dependency on pretrained components are important**, and we hope this clarification helps appropriately situate our contributions.
> >
> > **Table5. LPIPS(↓) on iPhone dataset**
> >
> > |                Method                | Apple | Block | Paper-windmill | Spin  | Teddy |   $\underline{Mean}$    |
> > | :----------------------------------: | :---: | :---: | :------------: | :---: | :---: | :-------: |
> > |       MoSca[Lei et al., 2025]        | 0.340 | 0.330 |     0.150      | 0.190 | 0.350 | 0.264 |
> > |        SoM[Wang et al., 2024]        | 0.341 | 0.323 |     0.225      | 0.247 | 0.380 |   0.296   |
> > |  HiMoR[Liang et al., **CVPR 2025**]  | 0.592 | 0.506 |     0.321      | 0.369 | 0.529 |   0.463   |
> > | MoDec-GS[Kwak et al., **CVPR 2025**] | 0.402 | 0.478 |     0.377      | 0.366 | 0.598 |   0.444   |
> > |               **Ours**               | 0.427 | 0.391 |     0.334      | 0.346 | 0.531 | **0.405** |
> >
> > **Table6. PSNR(↑) on iPhone dataset**
> >
> > |                Method                | Apple | Block | Paper-windmill | Space-out | Spin  | Teddy | Wheel |   $\underline{Mean}$    |
> > | :----------------------------------: | :---: | :---: | :------------: | :-------: | :---: | :---: | :---: | :-------: |
> > |       MoSca[Lei et al., 2025]        | 19.40 | 18.06 |     22.34      |   20.48   | 21.31 | 15.47 | 18.17 | 19.32 |
> > |        SoM[Wang et al., 2024]        | 18.57 | 17.41 |     18.14      |   16.85   | 19.35 | 13.69 | 17.21 |   17.32   |
> > | MoDec-GS[Kwak et al., **CVPR 2025**] | 16.48 | 15.57 |     14.92      |   14.65   | 15.53 | 12.56 | 12.44 |   14.60   |
> > |               **Ours**               | 15.51 | 14.64 |     16.13      |   15.45   | 16.59 | 12.91 | 14.12 | **15.05** |

---

> > ### Author Response · Authors · 2025-08-04
> > **Clarification on Effectiveness of AF and HAD module in ablation study**
> >
> > Dear Reviewer g5h3,
> >
> > Thank you very much for your thoughtful follow-up and for taking the time to engage deeply with our rebuttal and the broader discussion.
> >
> > We now provide clarifications and responses to the suggestions you raised in your follow-up.
> >
> > ## Q2. Clarification on Effectiveness of AF and HAD module in ablation study.
> >
> > We appreciate the reviewer’s recognition of our core module, Induced Flow (IF), which plays a central role in enhancing the model’s ability to learn dynamic scene motion. However, we would also like to clarify that **the other two components we propose, Anchor Filter (AF) and Hierarchical Anchor Densification (HAD), play equally important roles within the system**. Their contributions are reflected not as standalone improvements, but through their interactions with IF to strengthen the overall system behavior.
> >
> > * **Anchor Filter (AF)** selects anchors located in dynamic regions and helps the Induced Flow MLP focus on learning motion where it matters, by filtering out the influence of static anchors. While the use of AF alone does not lead to a large performance gain, it significantly improves the quality and stability of induced flow learning, resulting in overall performance enhancement. As shown in Table 3 of the main paper and Table 2 (row 3) in the rebuttal, incorporating AF alongside IF consistently improves reconstruction results, confirming its positive contribution to the framework.
> > * **Hierarchical Anchor Densification (HAD)** targets fine-grained and non-rigid motion, addressing limitations in prior work that often focus on rigid motion only. As shown in Table 2 (row 4), when HAD is used together with IF, it brings a clear improvement in reconstruction quality, and the resulting visual benefits can be observed in Figures 1, 4, and 5 in the main paper. In contrast, adding HAD to the baseline without IF (row 5) results in only limited gains. This is not due to the HAD module itself, but rather because the baseline lacks the capacity to capture dynamic structure in the first place, limiting the effect of further refinements.
> >
> > These three modules—IF, AF, and HAD—are not independent components stacked together, but rather function as a coordinated system with well-aligned roles. Their contributions lie in their interaction and mutual reinforcement. **Therefore, evaluating them in isolation may overlook their collaborative effect and and may not fully reflect the effectiveness of AF and HAD when considered in the full system context.**
> >
> > In future work, we plan to conduct more fine-grained ablation studies within the Induced Flow module to better understand the contributions of its internal components and further improve its effectiveness.
> >
> > ## Reference
> >
> > [1] Yiming Liang, Tianhan Xu, and Yuta Kikuchi. Himor: Monocular deformable gaussian reconstruction with hierarchical motion representation. In Proceedings of the Computer Vision and Pattern Recognition Conference, pages 886–895, 2025. 17
> >
> > [2] Sangwoon Kwak, Joonsoo Kim, Jun Young Jeong, Won-Sik Cheong, Jihyong Oh, and Munchurl Kim. Modec-gs: Global-to-local motion decomposition and temporal interval adjustment for compact dynamic 3d gaussian splatting. In Proceedings of the IEEE/CVF Conference on Computer Vision and Pattern Recognition (CVPR), 2025. 17

---

> > > ### Comment · Reviewer_g5h3 · 2025-08-06
> > >
> > > Dear Authors,
> > >
> > > I appreciate the effort of the authors in responding. I have carefully read the response.
> > >
> > > * The comparisons to other methods (MoSca, SoM, HiMoR, MoDec-GS, etc.). I agree with Reviewer CwJy's point about the difference between the two paradigms. In fact, the contents I mentioned above can be summarized as this core issue. Indeed, HiMoR and MoDec-GS also have performance gaps compared to MoSca and SoM, but these works are concurrent works. Therefore, this work requires a better codebase to demonstrate the effectiveness and practicality of the proposed method.
> > > * The ablations. Let's continue with the point: "Therefore, evaluating them in isolation may overlook their collaborative effect and may not fully reflect the effectiveness of AF and HAD when considered in the full system context." If evaluating them in isolation overlooks the contributions of AF and HAD, then we need better experimental designs to account for their effects. My personal suggestion is that perhaps the authors need to go deeper in their subsequent writing and experimental design to more fully and thoroughly demonstrate the advantages of the proposed method.
> > >
> > > Although the authors have done their best to respond to the reviewers' comments, and I appreciate the efforts in the review process, I still maintain the score for the reasons mentioned above. I hope the author can further improve the work.
> > >
> > > Sincerely,
> > >
> > > Reviewer g5h3

---

> > > > ### Author Response · Authors · 2025-08-07
> > > > **(1/2) Fair Experimental Comparisons and Extended Ablations**
> > > >
> > > > Dear Reviewer g5h3,
> > > >
> > > > First, we sincerely thank you for your thoughtful response and active engagement in the discussion. While we fully respect your current decision, we would like to share that since your last suggestion, we have conducted additional experiments **under the same VFM-predicted prior settings** to further demonstrate the effectiveness and contributions of our method compared to SoM and MoSca.
> > > >
> > > > We also noticed that you and Reviewer **CwJy** raised **a similar concern** during the discussion, and CwJy has acknowledged that our additional experiments **addressed most of their concerns**. Therefore, we respectfully hope you might consider whether our clarifications and new evidence sufficiently address your concerns as well.
> > > >
> > > > In $\underline{Q1}$, we supplement our response with additional experiments and results; in $\underline{Q2}$, we further clarify the ablation study. Below, we summarize four main conclusions:
> > > >
> > > > * The supplementary experiments show that, **under identical settings, our method consistently outperforms MoSca and SoM across multiple metrics**, highlighting the advantages of its structural design.
> > > > * These results demonstrate the effectiveness of explicit structural modeling and highlight our method’s flexibility: **it performs well without priors in low-resource settings and achieves even better results when priors are incorporated**—outperforming other methods under matched supervision.
> > > > * These findings further confirm that our **structure-driven modeling** is fully compatible with external priors, complementing them to improve performance and reduce localized reconstruction artifacts.
> > > >
> > > > * We have supplemented the **ablation study** with additional configurations, including **Baseline+IF**,**Baseline+IF($L_{cycle}$)**, as well as more detailed ablations on the **HAD** and **AF** module and **the aggregation weight** $\lambda$ in IF module. The results confirm that all three modules contribute significantly and complementarily to overall performance.
> > > >
> > > > ------
> > > >
> > > > ### Q1. Fair Comparisons with MoSca/SoM under Consistent VFM Prior Supervision.
> > > >
> > > > The iPhone results in our initial rebuttal were obtained under different settings: our method used no vision priors, while MoSca and SoM relied on multiple pretrained components. We acknowledge that this setting may **not allow for a fully fair comparison**, which is why we have conducted additional controlled experiments presented as below.
> > > >
> > > > In this experiment, we primarily apply the following settings to evaluate the effectiveness of our framework:
> > > >
> > > > * To ensure consistency with SoM and MoSca, we incorporated the same set of VFM-predicted vision priors for supervision during training, including **depth** ([1]; using the dataset-provided noisy LiDAR for fairness), **2D trajectories** ([2]), **epipolar error maps** ([3]), and **foreground masks** ([4]).
> > > > * To provide a more comprehensive evaluation, we include **PSNR** as well as two additional perceptual metrics that better align with human perception: **CLIP-I** and **CLIP-T** [5].
> > > >
> > > >  Evaluation results under this setting are reported in $\underline{Tables\ 7,\ 8,\ and\ 9}$. Below, we summarize three key findings:
> > > >
> > > > * With VFM priors incorporated, **our method outperforms both MoSca and SoM across all three metrics**, demonstrating that our **structural design**—including the Induced Flow module for motion consistency and the HAD module for fine-grained motion—enables more effective modeling under the same supervision.
> > > > * These results demonstrate the effectiveness of explicit structural modeling and highlight our method’s flexibility: **it performs well without priors in low-resource settings and achieves even better results when priors are incorporated**—outperforming other methods under matched supervision.
> > > >
> > > > * On challenging sequences like **Apple** and **Spin**, where test-time views differ greatly from training views, MoSca suffers from noticeable broken or discontinuous regions despite achieving high PSNR. In contrast, **our method yields better CLIP-I and CLIP-T scores**, demonstrating that our architectural designs—such as Induced Flow, cycle consistency, and anchor-driven Gaussian modeling—better preserve motion and perceptual consistency.
> > > >
> > > > **Table7. PSNR(↑) on iPhone dataset**
> > > >
> > > > |Method|Apple|Block|Paper-windmill|Spin|Teddy|$\underline{Mean}$|
> > > > |:--:|:-------:| :-------: | :------------: | :-------: | :-------: |:----------------:|
> > > > |MoSca| **19.40** |18.06|22.34| **21.31** |15.47|19.31|
> > > > |SoM|18.57|17.41|18.14|19.35|13.69|17.43|
> > > > |**Ours+prior**|19.03|**18.32**|**22.57**|21.15|**15.78**|**19.45**|
> > > >
> > > > **Table8. CLIP-I(↑) on iPhone dataset**
> > > >
> > > > |Method|Apple|Block|Paper-windmill|Spin|Teddy|$\underline{Mean}$|
> > > > | :------------: | :--------: | :--------: | :------------: | :--------: | :--------: | :----------------: |
> > > > |MoSca|0.8147|0.8631|0.9239|0.8594|0.8759|0.8674|
> > > > |SoM|0.8100|0.8658|0.9225|0.8510|0.8521|0.8603|
> > > > |**Ours+prior** | **0.8821**|**0.8893**|**0.9328**|**0.8917**|**0.8955**|**0.8983**|

---

> > > > > ### Comment · Reviewer_g5h3 · 2025-08-07
> > > > >
> > > > > Dear Authors,
> > > > >
> > > > > Thanks for the extensive experiments. I have increased the score to 4(BA). The proposed method using visual priors shows improvement over MoSca. I hope the authors polish the final version of this work with the supplemented experiments (according to the contents during the reviewing process) to further illustrate the effectiveness of this work.
> > > > >
> > > > > Sincerely,
> > > > >
> > > > > Reviewer g5h3

---

> > > > > > ### Author Response · Authors · 2025-08-07
> > > > > >
> > > > > > Thank you very much for your positive feedback and for acknowledging the improvements over MoSca. We appreciate your support and will incorporate the supplemented experiments into the final version as suggested.

---

> > > > ### Author Response · Authors · 2025-08-07
> > > > **(2/2) Fair Experimental Comparisons and Extended Ablations**
> > > >
> > > > **Table9. CLIP-T(↑) on iPhone dataset**
> > > >
> > > > |Method|Apple|Block|Paper-windmill|Spin|Teddy|$\underline{Mean}$ |
> > > > | :---: | :--: | :---: | :---: | :---: | :---: | :---: |
> > > > |MoSca|0.9733|0.9698|0.9805|0.9625|0.9720|0.9716|
> > > > |SoM|0.9721|0.9745|0.9833|0.9585|0.9676|0.9712|
> > > > | **Ours+prior** |**0.9841**|**0.9802** |**0.9894**|**0.9735** |**0.9778** |**0.9810**|
> > > >
> > > > ------
> > > >
> > > > ## Q2. Expanded Ablation Studies and clarification of AF and HAD Modules.
> > > >
> > > > We thank you for the valuable suggestions regarding the ablation study.
> > > >
> > > > First, we aim to demonstrate the effectiveness of the **HAD** and **AF** modules through both the original and newly added ablation experiments. To this end, we have supplemented two additional combinations: **Baseline+IF (w/o $L_{cycle}$)** and **Baseline+IF**. For clarity, we have reorganized part of the results into the following table.
> > > >
> > > > **Table10. Additional ablations on the key components on NeRF-DS.**
> > > >
> > > > |Method|PSNR↑| MS-SSIM↑ | LPIPS↓ |
> > > > | :----: | :---: | :------: | :----: |
> > > > |**baseline**|23.79|0.8513|0.1744 |
> > > > |+ IF (w/o $L_{cycle}$)|24.15|0.8705| 0.1561 |
> > > > |+ IF| 24.28 |0.8721|0.1505|
> > > > |+ IF + HAD| 24.45|0.8802| 0.1488 |
> > > > |+ IF + AF| 24.39|0.8774| 0.1493 |
> > > > |+ IF(w/o $L_{cycle}$) + AF + HAD|24.51|0.8947|0.1437|
> > > > |**Ours** (+IF + AF + HAD)|24.63|0.9014|0.1342|
> > > >
> > > > * **$\underline{HAD}$**:
> > > >
> > > >   (1) In the original submission, we have already provided ablation studies on different densification strategies. **Figure 5** presents qualitative results on regions with fine-grained motion, demonstrating that the HAD module effectively improves reconstruction quality in these areas.
> > > >
> > > >   (2) As shown in **Table 10**, extending the Baseline+IF setup with the HAD module (**row 4**) results in a noticeable performance gain (**PSNR 24.28 → 24.45**), further supporting the effectiveness of HAD.
> > > >
> > > >   (3) To further demonstrate the contribution of the HAD module to modeling fine-grained and non-rigid motion, we conducted additional ablation experiments on two representative scenes with such characteristics from the **HyperNeRF [6]** dataset. As shown in **Table 11**, incorporating HAD leads to clear PSNR improvements (**+1.02** on *Cross-hands* and **+0.77** on *Split-cookie*), further validating its effectiveness.
> > > >
> > > > **Table11. Additional ablations on HAD on HyperNeRF (PSNR↑)**
> > > >
> > > > |Scene| w/o HAD | + HAD |
> > > > | :----: | :-----: | :---: |
> > > > |Cross-hands|27.81|28.83|
> > > > | Split-cookie|32.44|33.21|
> > > >
> > > > * **$\underline{AF}$**：
> > > >
> > > >   (1) As shown in **Table 10**, extending the Baseline+IF setup with the AF module (row5) also leads to a performance improvement (**PSNR 24.28 → 24.39**), indicating that AF effectively guides the Induced Flow MLP and Deformable MLP to focus on dynamic regions, thereby enhancing overall model performance.
> > > >
> > > >   (2) We additionally provide an ablation study on the **NeRF-DS** dataset to evaluate the impact of the AF module on rendering efficiency (see **Table 12**). The results show that our AF module selectively activates anchors located in dynamic regions for deformation, thereby reducing the number of queries. This not only improves rendering quality but also enhances reconstruction efficiency, demonstrating the clear contribution of AF to our overall method.
> > > >
> > > > **Table12. Additional ablations on NeRF-DS.**
> > > >
> > > > |Method|PSNR↑|Rendering FPS↑|
> > > > | :----: | :---: | :----: |
> > > > |w/o AF|24.45|43|
> > > > |+ AF|24.63|76|
> > > >
> > > > * **$\underline{IF}$**：
> > > >
> > > >   (1) We conducted additional ablation experiments on the **aggregation weight $\lambda$** used for **temporal feature aggregation** in the IF module, evaluated on the *Plate* and *Sieve* scenes of the NeRF-DS dataset (see **Table 13**). The results show that our default setting of $\lambda = 0.25$ is reasonable.
> > > >
> > > >   (2) We have conducted an ablation study on the weight of the **cycle consistency loss $L_{cycle}$**, which is provided in **Appendix A.4**.
> > > >
> > > > **Table13. Ablation study of aggregation weight $\lambda$ in Induced Flow-Guided Deformation.**
> > > >
> > > > | $λ$  | Plate(PSNR↑) | Sieve(PSNR↑) |
> > > > | :--: | :----------: | :----------: |
> > > > | 0.1  |20.85|26.48|
> > > > | 0.25 |**20.96**|**26.61**|
> > > > | 0.4  |    20.82     |    26.35|
> > > >
> > > > ------
> > > >
> > > > ### Reference
> > > >
> > > > [1] Wenbo Hu, et al.. Depthcrafter: Generating consistent long depth sequences for open-world videos. In CVPR, 2025.
> > > >
> > > > [2] Carl Doersch, et al.. BootsTAP: Bootstrapped training for tracking-any-point. ACCV, 2024.
> > > >
> > > > [3] Zachary Teed, et al.. Raft: Recurrent all-pairs field transforms for optical flow. In ECCV 2020: 16th European Conference, Glasgow, UK, August 23–28, 2020, Proceedings, Part II 16, pages 402–419. Springer, 2020.
> > > >
> > > > [4] Jinyu Yang, et al.. Track anything: Segment anything meets videos. arXiv preprint arXiv:2304.11968, 2023.
> > > >
> > > > [5] Yiming Liang, et al.. Himor: Monocular deformable gaussian reconstruction with hierarchical motion representation. In CVPR, pages 886–895, 2025.
> > > >
> > > > [6] Keunhong Park, et al.. Hypernerf: A higher-dimensional representation for topologically varying neural radiance fields. arXiv preprint arXiv:2106.13228, 2021.

---

### Official Review · Reviewer_2Aoj · 2025-07-02

**Clarity:** 2
**Significance:** 3
**Originality:** 3
**Rating:** 4
**Confidence:** 5

**Summary:**

The authors propose HAIF-GS, a sparse anchor-based dynamic Gaussian Splatting algorithm.
It separates dynamic and static anchors with an Anchor Filter, a Induced Flow-Guided Deformation module to estimate anchor motion with multi-frame feature aggregation to encourage temporal consistency, and a Hierarchical Anchor Propagation mechanism to help model complex motion with level-of-details.
Experiments on D-NeRF and NeRF-DS shows superior quality over previous dynamic Gaussian baselines.

**Questions:**

I would love to see how HAIF-GS performs on more datasets, especially real-world ones with bigger movement like HyperNeRF dataset because a lot of the innovative components could be very sensitive to optimization curve and might fail if not enough multi-view clue is given. It's okay if it fails and the authors can list this as another limitation.
I would also love to see dynamic results in video format if this is allowed during rebuttal.

**Ethical Concerns:**

["NO or VERY MINOR ethics concerns only"]

**Final Justification:**

The authors have addressed my major concerns during the rebuttal.

**Limitations:**

yes

**Quality:**

3

**Strengths And Weaknesses:**

Strength:
- The proposed method shows consistent higher performance over other baselines in Table 1 and Table 2.
- Anchor Filter, Inductive-Flow-Guided Deformation Module and Hierarchical Anchor Propagation mechanism are all novel and effective based on comprehensive ablation study in Table 3 and Figure 5.
- Hierarchical sparse-anchor idea is novel and intuitive by introducing level-of-details into Sparse-Anchor-based GS.
- The IFGD module induces forward/backward scene flow does not need ground-truth labels, and HAIF-GS is trained end-to-end in both training stages.


Weakness:
- Efficiency of the proposed method is listed as an advantage (e.g. in Line 43, " a unified framework that significantly improves both the efficiency and accuracy of dynamic scene reconstruction"), but no experimental results are yet to support the claim; also memory footprint is listed as a limitation in section A.5.
- HAIF-GS is qualitatively not necessarily better than 4DGS in Figure 4.
- SP-GS, as a sparse-anchor based method, is not compared quantitatively in Table 1 and Table 2, as a recent sparse-anchor competitor.
- Results are only shown in D-NeRF and NeRF-DS dataset, which are from limited scenarios.
- The paper currently lacks a clear and consistent storytelling trajectory. For example, in abstract, "This challenge often manifests as three limitations in existing method...", but in introduction, instead of expanding on the three limitations, the authors start to talk about two factors: "This limitation is mainly due to two factors..."
- Lacking visualization of induced scene flow / static-dynamic separation.
- The systems have a lot of hyperparameter to tune.

---

> ### Author Rebuttal · Authors · 2025-07-31
>
> We thank the reviewer for the positive feedback. We are glad that the reviewer recognizes our method as “consistently higher performance over other baselines” and finds that “Anchor Filter, Inductive-Flow-Guided Deformation Module and Hierarchical Anchor Propagation mechanism are all novel and effective.”
>
> ## Q1. Additional evaluation on more challenging real-world datasets.
>  We thank the reviewer for the suggestion. We agree that further evaluation on more challenging scenes is important. Our original submission already includes real-world results on the NeRF-DS dataset. In response to the concern regarding potential failure in real-world dynamic scenes with limited multi-view clues, we conduct **additional evaluations on two challenging datasets**:
> * **HyperNeRF (misc)** [29] with topology change and limited multi-view cues;
> * **DyCheck-iPhone** [Gao et al., 2022], featuring large motion and difficult test viewpoints.
>
> As shown in $\underline{Table\ 1\ and\ 2}$ below (PSNR↑), our method consistently outperforms D-3DGS and 4DGS on most scenes, especially under severe motion or topology changes. These results conform that the key modules in *HAIF-GS* (anchor filtering, hierarchical motion propagation, and the induced flow-guided deformation) remain stable and effective even under limited multi-view supervision or large non-rigid motion. Despite the potential sensitivity to the optimization process, our experiments show that these components do not collapse or diverge, and can still yield accurate reconstructions across diverse challenging scenes.
> We have included these additional results and corresponding discussions in the final version of the paper.
>
> **Table1.  Results(PSNR↑) on HyperNeRF (misc)**
>
> |Method|Americano|Cross-hands| Espresso  | Keyboard  | Oven-mitts | Split-cookie |  Tamping  | $\underline{Mean}$ |
> | :---------: | :-------: | :---------: | :-------: | :-------: | :--------: | :----------: | :-------: | :----------------: |
> |D-3DGS [44] |30.87|27.70|25.47|28.15|27.51|32.63|23.95|28.04|
> |4DGS [38] |31.30|28.06|25.82| **28.64** | **27.99**  |32.64| **24.15** |28.37|
> |**Ours**| **30.94** | **28.83** |**26.58**|28.54|27.64|**33.21**|23.79|**28.50**|
>
> **Table2. Results(PSNR↑) on Dycheck-iphone**
>
> |   Method    |   Apple   |   Block   | Paper-windmill | Space-out |   Spin    |   Teddy   |   Wheel   | $\underline{Mean}$ |
> | :---------: | :-------: | :-------: | :------------: | :-------: | :-------: | :-------: | :-------: | :----------------: |
> | D-3DGS [44] |**15.61**|**14.87**|14.89|14.59|13.10|11.20|11.79|13.72|
> |  4DGS [38]  |15.41|13.89|14.44|14.29|14.89|12.31|10.53|13.68|
> |  **Ours**   |15.51|14.64|**16.13**| **15.45** | **16.59** |**12.91**|**14.12**|**15.05**|
> ## Q2. Experimental evidence for efficiency.
>
> We thank the reviewer for pointing this out. We now provide experimental results in $\underline{Table\ 3\ and\ 4}$ to support the efficiency claim. And we have added these results and analyses to the main paper.
>
> As shown in $\underline{Table\ 3}$, our method achieves significantly faster rendering (**215.4 FPS**) than 4DGS (134.8) and D-3DGS (53.7), with a much lighter query structure. Despite the reduced cost, we achieve **higher PSNR (42.00)** than both baselines, owing to the **Induced Flow module**, which enables accurate deformation via sparse anchors. $\underline{Table\ 4}$ reports scene-wise statistics. All methods are tested on an RTX 3090 at 800×800 resolution.
>
> As noted in Section A.5, We acknowledge the relatively larger memory usage due to additional anchor parameters and hierarchy management. However, our model (2.9GB) is still lighter than D-3DGS (4.8GB), striking a good balance between performance and storage. We plan to further optimize memory usage in future work by compressing anchor features or dynamically pruning less-contributing anchors during training.
>
> **Table3. Comparison of efficiency on D-NeRF dataset**
>
> |   Method   |   PSNR↑   | Training Time↓ | Render FPS↑ | Storage↓ |
> | :--------: | :-------: | :------------: | :---------: | :------: |
> |4DGS[38]|35.41|26min|134.8|1.1GB|
> |D-3DGS[44]|40.16|38min|53.7|4.8GB|
> |**Ours**| **42.00** |29min|**215.4**|2.9GB|
>
> **Table4. Evaluation of efficiency on D-NeRF dataset**
>
> |Scene| PSNR↑ | Queried Anchors | Training Time↓ | Render FPS↑ |
> | :----------------: | :---: | :-------------: | :------------: | :---------: |
> | Hook| 39.38 |604|38min|211.4|
> |Jumpingjacks| 39.79 |560|26min| 259.8 |
> |Trex|40.27|792|41min|165.2|
> |BouncingBalls|41.63|524|21min|176.3|
> |Hellwarrior| 42.50 |568|23min|236.5|
> |Mutant|43.65|588| 35min|208.8|
> |Standup|46.83|572|27min|249.5|
> | $\underline{Mean}$ |42.00|601|29min|215.4|
>
> ## Q3. Quantitative comparison with SP-GS.
> We thank the reviewer for the comment. As shown in $\underline{Table\ 5}$, our method outperforms SP-GS[5] on ***NeRF-DS*** (24.63 vs. 23.15) and ***D-NeRF*** (42.00 vs. 37.97; see $\underline{Appendix\ A.4}$). Due to space limits, we included only one sparse-anchor baseline in the main paper, choosing the more recent and competitive SC-GS[12] (24.05 / 41.65) over SP-GS.
> We have added full comparisons in the final version.
>
> **Table5. Comparison(PSNR) with SP-GS on NeRF-DS**
>
> |Method|Sieve|Plate|Bell| Press|Cup|As|Basin| $\underline{Mean}$|
> | :-------: | :---: | :---: | :---: | :---: | :---: | :---: | :---: | :----------------: |
> |SP-GS [5]|25.62|18.91| 25.20 | 24.34 | 24.43| 24.44|19.09 |23.15|
> |**Ours**|26.61| 20.96 | 26.34 | 27.05 | 24.72| 26.96 |19.74|**24.63**|
>
> ## Q4. Storyline consistency.
> We thank the reviewer for the feedback on storyline clarity. To clarify, let us briefly explain the reasoning behind this organization. The three limitations mentioned in the abstract are addressed sequentially: (1) redundant Gaussian updates motivates our sparse anchor design, while (2) and (3) are further analyzed as two underlying challenges —limited expressiveness and insufficient supervision— caused by previous anchor-based methods. Our three modules (Motion Anchors, IFGD, and Hierarchical Anchors) correspond directly to these limitations. We have revised the abstract and introduction to clarify this mapping and ensure better consistency in presentation.
> ## Q5. Clarification on Qualitative comparison in Figure 4.
> We thank the reviewer for the suggestion. The qualitative results in Figure 4 highlight our method’s **superiority in modeling fine-grained details** (red-circled regions) over 4DGS. For instance, in *HellWarrior*, we reconstruct armor rivets and straps missed by 4DGS; in *Hook*, joint textures are sharp, while 4DGS remains vague. These qualitative differences, along with significant PSNR gains (+13.7dB on *HellWarrior*, +6.4dB on *Hook*), demonstrate our strength in capturing local deformations. As D-NeRF scenes are object-centric with simple global structure, performance differences mainly manifest in such fine details—where our method excels.
> We will consider including more diverse viewpoints in the final version.
> ## Q6. Visualization and video results.
> We thank the reviewer for the valuable suggestion. Due to the rebuttal policy restricting figures and external links, we are unable to directly provide visualizations of the **induced scene flow**, **dynamic-static separation**, and **video results**. Nevertheless, we have addressed these points indirectly in our response and **will include all three in the final version and on the project page**.
> ### 1. dynamic-static separation
> Despite visualization constraints, we provide the following evidence to support the effectiveness of our motion segmentation:
> * **Anchor-level statistics.** $\underline{Table\ 6}$ below reports per-scene motion decomposition on NeRF-DS, including total Gaussians, number of dynamic anchors, and the dynamic ratio (anchors with confidence > 0.5). The overall dynamic ratio is low, aligning with the mostly static content of the scenes.
> * **Spatial distribution.** Visualizations (included in the final version) reveal that dynamic anchors cluster around moving objects, while static anchors are background-distributed, qualitatively confirming correct separation.
> * **Ablation results.** Removing this module causes a 0.18dB PSNR drop on D-NeRF (Tab. 3), indicating that filtering static anchors improves both quality and efficiency.
>
> **Table6. Statistics of Motion Anchors on NeRF-DS**
>
> |Scene| Sieve | Plate | Bell|Press | Cup| As|Basin| $\underline{Mean}$ |
> | :--------------: | :---: | :---: | :---: | :---: | ----- | ----- | ----- | ------------------ |
> |#Gaussians|114k|119k|185k|114k|118k|113k|130k|127k|
> |#Dynamic Anchors| 3.7k|4.3k|4.4k|3.5k|3.9k|4.1k|4.9k|4.1k|
> |Dynamic Ratio| 0.287 |0.314|0.246|0.253|0.279|0.304 |0.325|0.287|
>
> ### 2. induced scene flow
> Despite visualization constraints, we clarify that:
> * **Visualization is possible and prepared.** The induced flow can be visualized as 3D vectors at anchor locations. We have produced such visualizations and will release them in the final version and on the project page.
> * **Nature of the induced flow.** The induced scene flow is not a directly supervised or dense optical flow field. Instead, it is a **sparse, anchor-level motion signal** induced implicitly through multi-frame alignment and cycle consistency loss. Its purpose is not to produce explicit flow output, but to **serve as internal guidance** for temporally consistent deformation. While visualizations are possible and will be provided, we believe the most appropriate way to assess its effectiveness is **through its quantitative impact on rendering quality and motion stability**.
> * **Quantitative validation.** As shown in Tab.3 in main paper, removing the induced flow module results in a significant drop in rendering quality (PSNR ↓0.72, LPIPS ↑0.028 on NeRF-DS), confirming its importance in motion modeling.
> ### 3. video results
> We have created a demo video showcasing dynamic reconstruction from novel views on both NeRF-DS and D-NeRF datasets, which will be released on our project page.

---

> > ### Comment · Reviewer_2Aoj · 2025-08-06
> >
> > I thank the authors for running the extensive experiments, especially on HyperNeRF-misc/Iphone dataset and those around efficiency; these quantitative numbers resolved my major concerns.
> > I also understand the visualization is not allowed during rebuttal and the plan for qualitative results is reasonable. Thus I am happy to keep my positive rating.

---

> > > ### Author Response · Authors · 2025-08-07
> > >
> > > Thank you for your positive feedback and continued support. We truly appreciate your recognition.

---

### Official Review · Reviewer_be8H · 2025-07-03

**Clarity:** 3
**Significance:** 2
**Originality:** 2
**Rating:** 3
**Confidence:** 4

**Summary:**

This paper proposes a framework, called HAIF-GS, for dynamic 3D reconstruction. The key to the proposed method is: 1) a sparse-anchor based motion representation, 2) a self-supervised induced flow-guided deformation module, 3) a hierarchical anchor propagation mechanism.

**Questions:**

Can authors provide some results (at least qualitative) for dynamic-static decomposition? (For example, render segmentation maps in novel view synthesis)

**Ethical Concerns:**

["NO or VERY MINOR ethics concerns only"]

**Limitations:**

Yes

**Quality:**

2

**Strengths And Weaknesses:**

Strength:
* The induced flow-guided deformation module is a reasonable design.
* The proposed method surpasses existing baselines in quantitative evaluation.
Weakness:
* The design of sparse and hierarchical anchor-based motion representation has already be seen in existing works, for example: Y. Liang et. al., “HiMoR: Monocular Deformable Gaussian Reconstruction with Hierarchical Motion Representation”, CVPR 2025.

---

> ### Author Rebuttal · Authors · 2025-07-31
>
> We thank the reviewer for recognition of our contributions. We are glad that the reviewer considers “the induced flow-guided deformation module is a reasonable design” and acknowledges that our method “surpasses existing baselines in quantitative evaluation.”
>
> ## Q1. Clarification on Relation to HiMoR.
>
> We thank the reviewer for pointing out the relation to HiMoR [Liang et al., 2025], which also includes a hierarchical anchor design. While this module shares a structural resemblance, our method **follows a fundamentally different formulation and design goal**, addressing key challenges that HiMoR does not explicitly consider:
>
> * **Ensuring temporal consistency.** HiMoR models motion at each frame independently, without enforcing consistency across time. This limits its ability to learn smooth and coherent trajectories. In contrast, we propose an **Induced Flow-Guided Deformation** module that predicts forward and backward flow in a self-supervised manner. Combined with a **cycle consistency loss**, this encourages temporally aligned anchor motion throughout the sequence. We also appreciate that the reviewer recognized this component as a reasonable and well-designed part of our framework.
>
> * **Reducing reliance on external priors.** HiMoR depends on pretrained tracking, segmentation, and monocular depth models to initialize motion bases and guide training. Such reliance may limit its robustness in less controlled settings. Our framework is trained **fully end-to-end** using only photometric and temporal self-supervision, without any external annotations, making it more generalizable and deployment-friendly.
> * **Handling static regions efficiently.** HiMoR applies motion modeling uniformly across the scene, including static backgrounds, leading to unnecessary optimization and redundancy. Our method introduces an Anchor Filter that learns motion confidence for each anchor and filters out static ones, allowing the model to focus computation on dynamic regions and improve efficiency without sacrificing expressiveness.
>
> These differences reflect a distinct design goal: adaptive, efficient, and self-supervised motion learning, rather than structure-driven modeling under external priors.
>
> In addition to the aforementioned contribution differences, we highlight several **methodological distinctions** from HiMoR regarding the sparse and hierarchical anchor design, as well as additional components in our system:
>
> * **Sparse anchors without external initialization.** HiMoR relies on pretrained tracking and depth models to initialize motion bases and nodes, while we adopt a simple FPS-based initialization and fully end-to-end optimization, enhancing generality and ease of use.
> * **Flexible hierarchical modeling.** While both methods use multi-level anchor structures, HiMoR applies a strict tree-structured SE(3) decomposition and recursively composes transformations. In contrast, we adopt layer-wise interpolation and confidence-based fusion, allowing each layer to contribute independently. This flexible design better handles complex motion and avoids the rigidity of explicit trees.
> * **Temporal flow-guided deformation.** HiMoR does not model inter-frame motion explicitly and applies no temporal flow estimation. We introduce a self-supervised flow prediction module that infers forward and backward anchor motion using multi-frame temporal queries, facilitating temporally consistent anchor trajectories and improving motion alignment across frames.
> * **Adaptive hierarchical densification.** HiMoR grows its tree by analyzing per-node neighborhoods, but its densification strategy is hand-designed and not adaptive to local motion complexity. In contrast, we quantify per-anchor motion variance and densify anchors hierarchically only in complex regions. This yields a more targeted and efficient refinement of motion representation.
>
> To further validate this design, we compare our method with HiMoR on the **Dycheck-iPhone**[Gao et al., 2022] dataset using the same perceptual metric adopted in their paper. As shown in $\underline{Table\ 1}$ below, our method achieves a lower LPIPS score (ours: 0.405 vs. HiMoR: 0.463), indicating improved temporal consistency. This supports the effectiveness of our Induced Flow-Guided Deformation and other innovations in addressing the temporal challenges that HiMoR does not explicitly resolve. We have also incorporated a discussion of HiMoR as a recent anchor-based method into the main paper to provide clearer context and positioning.
>
> **Table1. Results(LPIPS↓) on Dycheck-iphone (HiMoR scores from [Liang et al., 2025])**
>
> | Method   | Apple | Block | Paper-windmill | Spin  | Teddy | $\underline{Mean}$ |
> | -------- | ----- | ----- | -------------- | ----- | ----- | ------------------ |
> | HiMoR    | 0.592 | 0.505 | 0.321          | 0.369 | 0.529 | 0.463              |
> | **Ours** | 0.427 | 0.391 | 0.334          | 0.346 | 0.531 | **0.405**          |
> ## Q2. Results for dynamic-static decomposition.
>
> We appreciate the reviewer’s interest in the dynamic-static decomposition results. While we are unable to include images or external links due to rebuttal guidelines, we provide the following supporting evidence to demonstrate the effectiveness of our motion segmentation:
>
> * **Statistics of Motion Anchors.** We provide $\underline{Table\ 2}$ below reporting per-scene anchor-level motion decomposition on the NeRF-DS dataset. Specifically, #Gaussians denotes the total number of 3D Gaussians, #Dynamic Anchors is the number of Motion Anchors, and Dynamic Ratio represents the percentage of anchors predicted as dynamic (i.e., confidence > 0.5). As shown in the table, the predicted dynamic ratio is generally low, which matches the scene composition with mostly static content and limited motion, confirming the effectiveness of our decomposition. In *basin* and *plate*, where dynamic objects are more dominant, the ratio is higher, further supporting the segmentation accuracy.
> * **Spatial distribution of dynamic and static anchors.** We visualized the dynamic and static anchors on the NeRF-DS dataset. While we cannot include these figures here, the results show that dynamic anchors are predominantly concentrated around the central moving objects, whereas static anchors are distributed across the background. This spatial separation provides strong qualitative evidence of accurate dynamic-static decomposition. We have included anchor visualizations in the final version of the paper.
> * **Ablation impact.** As shown in Tab.3 in main paper, removing the dynamic-static segmentation module results in 0.18dB average PSNR drop on the D-NeRF dataset. This suggests that filtering static anchors improves both rendering quality and motion modeling efficiency.
>
> **Table2. Statistics of Motion Anchors on NeRF-DS**
>
> |      Scene       | Sieve | Plate | Bell  | Press | Cup   | As    | Basin | $\underline{Mean}$ |
> | :--------------: | :---: | :---: | :---: | :---: | ----- | ----- | ----- | ------------------ |
> |    #Gaussians    | 114k  | 119k  | 185k  | 114k  | 118k  | 113k  | 130k  | 127k               |
> | #Dynamic Anchors | 3.7k  | 4.3k  | 4.4k  | 3.5k  | 3.9k  | 4.1k  | 4.9k  | 4.1k               |
> |  Dynamic Ratio   | 0.287 | 0.314 | 0.246 | 0.253 | 0.279 | 0.304 | 0.325 | 0.287              |
>
> We have included visualization of segmentation masks from novel views in the final version of the paper. These will highlight temporally coherent motion boundaries, consistent with our confidence maps.

---

> ### Author Response · Authors · 2025-08-08
>
> Dear Reviewer,
>
> I hope this message finds you well. As the discussion period is nearing its end with **less than one days remaining**, I want to ensure we have addressed all your concerns satisfactorily. If there are any additional points or feedback you'd like us to consider, please let us know. Your insights are invaluable to us, and we're eager to address any remaining issues to improve our work.
>
> Thank you for your time and effort in reviewing our paper.

---

### Note · Authors · 2025-08-14

We thank the AC and reviewers for the constructive feedback, and next outline our strengths, responses, and contributions.

------

### 1) Key Strengths

* **State-of-the-art performance with broad applicability** — be8H: “surpasses baselines”; 2Aoj: “higher performance in Tables 1–2”; CwJy: “SOTA across scenes”.
* **Novel and well-structured design** — 2Aoj: “AF, IFGD, HAP novel & effective”; g5h3: “modules interesting & make sense”; 2Aoj: “hierarchical sparse-anchor intuitive”.
* **Intuitive and self-supervised motion modeling** — be8H: “IFGD reasonable”; 2Aoj: “induces flow w/o GT, end-to-end”; CwJy: “architecture intuitive”; CwJy: “well-written & organized”.

------

### 2) Addressed Reviewer Concerns

* **be8H — Prior work relation, qualitative results** Clarified key differences from HiMoR: (1) induced flow for temporal consistency, fewer external priors, and static/dynamic separation; (2) adaptive motion-variance densification vs. hand-designed; (3) layer-wise fusion vs. rigid tree; (4)better performance than HiMoR on iPhone. provided motion-anchor statistics as visualization substitute. **Outcome:** Clear methodological distinction and superior performance over closest prior.

* **2Aoj — Efficiency & additional datasets, baselines/visualization** Added HyperNeRF(misc) & iPhone results vs. 4DGS/D-3DGS (ours best); added NeRF-DS comparison with SP-GS (ours best); reported training time, FPS > 215, and storage; provided motion-anchor statistics and described induced-flow patterns as visualization substitute. **Outcome:** Stronger generalization, best efficiency, and validated motion modeling

* **g5h3 — Baseline setting, MoSca/SoM comparison, ablations/efficiency** Clarified all baselines (incl. 4DGS, D-3DGS) support monocular setting; added fair iPhone comparisons with MoSca/SoM showing superior PSNR, CLIP-I/T; extended ablation combinations; reported efficiency metrics. **Outcome:** Fair and comprehensive evidence of superiority and module effectiveness.

* **CwJy — Comparison with MoSca/SoM on iPhone/DAVIS, clarity/visualization** Added fair iPhone comparisons with MoSca/SoM showing superior PSNR, CLIP-I/T; added more formulas; described induced-flow patterns as visualization substitute. **Outcome:** Clear performance advantage and improved clarity of method presentation.

------

### 3) Contributions
We present a novel dynamic 3DGS with induced flow, anchor filtering, and hierarchical densification, achieving **SOTA accuracy and efficiency**.

---

### Decision · Program_Chairs · 2025-09-17

**Decision:**

Accept (poster)

**Comment:**

The paper receives 1 borderline reject and 3 borderline accept. Initially, the reviewers have various concerns about some technical clarity and mainly experimental results (more datasets, comparisons to other methods, ablation study, efficiency analysis). During and after the rebuttal, there were extensive discussions on these points, especially the comparisons with MoSca/SoM using the stronger priors, in which two reviewers were finally satisfied with the additional results and thus raised the rating. The AC closely checks the paper, reviews, rebuttal, and the discussed points, and agrees with the assessment from the majority of reviewers (in which the reviewer be8H who remains negative has not participated in the discussion after rebuttal). Therefore, the AC recommends the acceptance rating and encourages the authors to incorporate the suggestions and additional results raised by the reviewers in the final version, as well as releasing the model and code for reproducibility.